



# Advances in the True Eddy Accumulation Method: New theory, implementation, and field results

Anas Emad[1] and Lukas Siebicke[1]

[1]Bioclimatology, University of Göttingen, Büsgenweg 2, 37077 Göttingen, Germany

**Correspondence:** Anas Emad (anas.emad@stud.uni-goettingen.de)

**Abstract.** The true eddy accumulation method (TEA) provides direct measurements of ecosystem-level fluxes for a wide range of atmospheric constituents. TEA utilizes conditional sampling to overcome the requirement for a fast sensor response usually demanded by the state-of-the-art eddy covariance method (EC).

However, the assumptions and conditions required for the TEA method are often not met. Here we explore the limitations

caused by the assumption of zero mean vertical wind velocity during the averaging interval and by the fixed accumulation interval.

We extend the theory of TEA method to non-zero vertical wind velocity by employing information about the scalar transport. We further derive a new method with adaptive time varying accumulation intervals. The new method, termed short-time eddy accumulation (STEA), was successfully implemented and deployed to measure $CO_2$ fluxes over an agricultural field in Braun-

schweig, Germany. The measured fluxes matched very well against a conventional EC system (slope of 1.05, $R^2$ of 0.87). We provide a detailed description of the setup and operation of the STEA system in the flow-through mode, devise an empirical correction for the effect of buffer volumes, and describe the important considerations for the successful operation of the STEA method.

The new theory developments reduce the bias and uncertainty in the measured fluxes and create new ways to design eddy

accumulation systems with finer control over sampling and accumulation. The results encourage the application of TEA and STEA for measuring fluxes of more challenging atmospheric constituents such as reactive species as well as other constituents where no fast gas analyzers are available.

## 1 Introduction

Micrometeorological methods provide non-invasive, in situ, integrated, and continuous point measurement for ecosystem fluxes

on a scale ideal for ecosystem study (Baldocchi et al., 1988; Baldocchi, 2014). Among micrometeorological methods, eddy covariance (EC) has become the de-facto method for measuring ecosystem fluxes for the past 40 years. The EC method is the most direct micrometeorological method. It is also relatively easy to set up and operate. These features have led to the wide use and adoption of the EC method at hundreds of sites worldwide, including several regional and global flux measurements networks such as ICOS and FLUXNET (Hicks and Baldocchi, 2020).





The EC method depends on the fast measurement of vertical wind velocity and the scalar concentration (such as an atmospheric constituent). The requirement for fast measurement frequency (10 to 20 Hz) limits the application of the method to a handful of atmospheric constituents where fast gas analyzers are available. Alternative methods that work for slow gas analyzers include: (i) signal downsampling methods (Lenschow et al., 1994) such as disjunct eddy accumulation (Rinne et al., 2000a; Turnipseed et al., 2009) and disjunct eddy covariance (Rinne and Ammann, 2012), and (ii) indirect methods such as flux

gradient methods (e.g Rinne et al. (2000b)) which depend the Monin-Obukhov similarity theory (Monin and Obukhov, 1959) and the relaxed eddy accumulation (REA) based on the assumption of flux-variance similarity (Businger and Oncley, 1990) The true eddy accumulation method (TEA) (Desjardins, 1977) is the most direct and mathematically equivalent alternative to eddy covariance. Unlike EC, the TEA method requires the concentration measurements to be carried out once every averaging interval (30 minutes). For a long time, the development of the TEA method was hindered by the difficulty of fast air flow rate

control and the strict operational requirements (Businger and Oncley, 1990; Hicks and McMillen, 1984). A recent improvement to the TEA method used a new type of mass flow controllers, online coordinates rotation, and several online treatments of the signal to overcome important limitations of the method's applicability (Siebicke and Emad, 2019). The new system showed a good match with a reference eddy covariance system with coefficients of determination of up to 86% and a slope of 0.98. The non-necessity of high-frequency measurements of the scalar in TEA, although a major advantage, introduces multiple

difficulties. First and foremost, exact equality between EC fluxes and TEA fluxes is only possible when the mean vertical wind velocity is assumed to be zero during the flux averaging interval. This assumption is almost never met under field conditions and the residual vertical mean velocity contributes to the error in the flux. Nonzero mean vertical wind velocity is a source of error for all eddy accumulation methods, including TEA (Hicks and McMillen, 1984), relaxed eddy accumulation (REA) (Pattey et al., 1993; Businger and Oncley, 1990; Bowling et al., 1998), and disjunct eddy accumulation (DEA) (Turnipseed et al.,

2009). The reported bias on the flux due to nonzero $\bar{w}$ varied with different studies and accumulation methods. For TEA, Hicks and McMillen (1984) recommended that $\bar{w}$ should not exceed $0.0005\,\sigma_w$ if accumulated mass is measured and $0.02\,\sigma_w$ when concentrations are measured directly. Turnipseed et al. (2009) reported that a mean vertical wind bias of $\pm\,0.25\,\sigma_w$ lead to $\pm\,15\%$ errors in the flux using the disjunct eddy accumulation method. Values reported for the REA method showed that a loss of approximately 5% of the flux due to a $\bar{w}$ of $0.20\,\sigma_w$ (Pattey et al., 1993), which agrees with the recommendations

of Businger and Oncley (1990). Additionally, the absence of high-frequency information means any decision on air sampling such as flow rate and sampling direction is final. The lack of high-frequency information also implies that sample accumulation should happen on a time scale similar to the flux averaging interval (30 to 60 minutes). These limitations impose restrictive design considerations related to the size and function of sample accumulation reservoirs. They also dictate that the sampling apparatus needs to accommodate a large dynamic range to cover the range of wind velocities during the flux averaging interval.

In this paper, we address these limitations of the TEA method. First, we revise the theory of true eddy accumulation and extend the TEA equation to non-ideal conditions. The new generalized TEA equation allows obtaining TEA fluxes equivalent to fluxes measured with EC when the vertical wind velocity is nonzero. This is achieved by incorporating knowledge about the scalar transport represented by the transport asymmetry coefficient. We show analytical and empirical ways to obtain the transport asymmetry coefficient and provide an interpretation of its value. Then, we describe the sensitivity of the flux to values





of the transport asymmetry coefficient, the residual vertical mean velocity, and the different operational conditions. Next, we derived a new TEA method, the short-time eddy accumulation (STEA), which allows to carry out the sample accumulation on variable intervals shorter than the flux averaging interval. We discuss the advantages and steps required to carry out flux measurements using the STEA method, different operational requirements, and develop an empirical correction for the use of buffer volumes. Finally, we show a prototype and experimental measurements for $CO_2$ fluxes using the newly developed

STEA method in flow-through mode and compare the measured fluxes to reference EC measurements

## 2 Theory

### 2.1 Eddy covariance

The net ecosystem exchange (NEE) of a scalar $c$ (such as an atmospheric constituent), $\overline{N_c}$ is the total vertical flux $\overline{wc}$ across the measurement plane at a height $h$ and the change of storage below that height (Gu et al., 2012).

$$\overline{N_c} = \overline{wc}\big|_h + \int_0^h \frac{\partial c}{\partial t} \mathrm{d}z \qquad (1)$$

Where $w$ is the vertical wind velocity ($\mathrm{m\,s^{-1}}$), $c$ is the molar density ($\mathrm{mol\,m^{-3}}$) of the scalar of interest (such as $CO_2$). The previous equation can be reached either from a holistic mass balance approach or by averaging the continuity equation for the scalar $c$ and integrating from the surface to measurement height $h$. In both cases, horizontal advection is ignored as a virtue of the assumption of horizontal homogeneity and molecular diffusion is ignored due to its small magnitude (Gu et al., 2012). For

a full discussion on the equations of surface flux, see, for example (Finnigan et al., 2003; Foken et al., 2012a).

The storage term measurements and value are beyond the scope of this study, therefore we ignore it. Consequently, the total vertical flux is represented by the first term on the right hand side of Eq. (1) which can be further decomposed into turbulent and mean advective parts.

$$\overline{wc} = \overline{w'c'} + \overline{w}\,\overline{c} \qquad (2)$$

The overlines denote ensemble averages that obey Reynolds averaging rules. Primes represent departures from the mean. The ensemble averages are estimated experimentally by the time averages, thus for a stationary time series drawn from an ensemble, the flux for the averaging period $T_{avg}$ can be written as

$$F_{T_{avg}} = \frac{1}{T_{avg}} \int_0^{T_{avg}} w'(t)c'(t)\, dt = \overline{w'c'} \qquad (3)$$

where $w(t)$ and $c(t)$ are realizations of the vertical wind velocity and the scalar quantity such as $CO_2$ concentration, respec-

tively.





The first term on the right-hand side of Eq. (2) is the covariance between vertical wind velocity and the scalar concentration. It is often referred to as the turbulent flux or the first-order approximation of the eddy flux, whereas the mean vertical advective term (second term on the right-hand side of Eq. (2)) is known as "Webb correction" or Webb-Pearman-Leuning (WPL) correction (Webb et al., 1980), and exists due to fluctuations in air density (Fuehrer and Friehe, 2002). The mean vertical wind velocity induced by density fluctuations of air parcels, hereafter $\overline{w_d}$, is often nonzero and is needed to account for the mean advective term in the flux. It is, however, not possible to directly measure it. One reason is that $\overline{w_d}$ caused by air density fluctuations is rather small (less than $1 \mathrm{~mm\,s^{-1}}$). But more importantly, any offset in the vertical wind velocity will appear in the measured $\overline{w}$, consequently obscuring $\overline{w_d}$. Several reasons can contribute to a nonzero mean vertical wind velocity. This includes: tilted coordinates, biases in instruments, flow perturbations, and meteorological reasons induced by local circulation or topographical drainage (Lee et al., 2005; Paw U et al., 2000; Heinesch et al., 2007). As a result, the measured slowly varying term $\overline{w}\,\overline{c}$ need to be discarded and the correct $\overline{w_d}$ need to be estimated. One way to estimate $\overline{w_d}$ is by utilizing the knowledge of the NEE of another scalar (Gu et al., 2012) or, in the case of WPL theory, by assuming that the net mean vertical mass flux of dry air is zero (stationarity of dry air) and calculating $\overline{w_d}$ from sensible and latent heat fluxes.

## 2.2 True Eddy Accumulation

The true eddy accumulation method circumvents the need to measure individual realizations of the scalar concentration. Instead, it is sufficient to measure the mean product $\overline{wc}$ for updraft and downdraft once at the end of each averaging interval $T_{avg}$.

The product of $w$ and $c$ is realized by physically collecting air samples with a flow rate proportional to the vertical wind velocity $w$. The method is formulated assuming ideal conditions, where the mean vertical wind velocity during the averaging period is assumed to be zero. When $\bar{w} = 0$, the second term on the right hand side of Eq. (2) will be zero and the turbulent flux $\overline{w'c'}$ will equal the mean product $\overline{wc}$. By separating $\overline{wc}$ depending on the direction of the vertical wind velocity we can write

$$\overline{wc} = \frac{1}{T_{avg}} \int\limits_0^{T_{avg}} (\delta^+ cw + \delta^- cw)\, \mathrm{d}t \tag{4}$$

$$\text{where} \quad \begin{cases} w > 0 & \delta + = 1;\ \delta - = 0 \\ w < 0 & \delta + = 0;\ \delta - = 1 \end{cases} \tag{5}$$

Hence, by sampling air with a flow rate proportional to vertical wind velocity and accumulating it according to its direction in updraft and downdraft reservoirs, one can measure the quantity $\overline{wc}$ and consequently the flux without having to measure the high-frequency fluctuations of the scalar, $c$ (Desjardins, 1977; Hicks and McMillen, 1984).

A simpler alternative formulation can be reached using the law of total expectation. We write the flux as the expected value of the random variable $wc$ conditional on the direction of the vertical wind velocity $\mathrm{sign}(w)$





$$\overline{wc} = \overline{\left((wc)|\mathrm{sign}(w)\right)} \tag{6}$$

Sampling air proportional to the magnitude of vertical wind velocity requires a scaling parameter, $A$ that maps vertical wind velocity to the flow rate. The scaling parameter is the product of the pump calibration coefficients and other coefficients used to adjust the system's dynamic range. For a short interval of time $\mathrm{d}t$ a sample of the volume $V_{sample} = A|w|\,\mathrm{d}t$ will be collected in the system.

The accumulated sample volume in each of the two reservoirs during a long enough averaging period $T_{avg}$ (30 to 60 minutes)
will be

$$V_{total} = \frac{1}{T_{avg}} \int\limits_{0}^{T_{avg}} A|w|\,\mathrm{d}t \tag{7}$$

By the end of the averaging period $T_{avg}$, the flux will be equal to the difference in the scalar accumulated mass between updraft and downdraft reservoirs.

If it is desired to formulate the flux in terms of the accumulated scalar concentration $(\mathrm{mol\,m^{-3}})$ instead of the accumulated
mass, the average density of accumulated samples in each of the reservoirs will equal the accumulated mass of the scalar divided by the accumulated volume

$$C_{acc}^{\uparrow\downarrow} = \frac{m}{V} = \frac{A \int_0^T c \,|\delta^{\pm} w|\,dt}{A \int_0^T |\delta^{\pm} w|\,dt} \tag{8}$$

Where $C_{acc}$ is the accumulated scalar density and the arrows indicate the reservoir. The measured concentration in Eq. (8) is the weighted mean of the scalar concentration and the magnitude of the vertical wind velocity. We can simply rewrite the
accumulated concentration in terms of the wind and the scalar concentration as

$$C_{acc}^{\uparrow\downarrow} = \frac{\overline{c\,|w^{\uparrow\downarrow}|}}{\overline{|w^{\uparrow\downarrow}|}} \tag{9}$$

When $\overline{w}$ is assumed to be zero, $\overline{|w^{\uparrow}|} = \overline{|w^{\downarrow}|} = \overline{|w|}/2$, and we can write the flux in terms of concentrations of accumulated samples, similar to Hicks and McMillen (1984)

$$F_{\mathrm{TEA}} = \frac{\overline{|w|}}{2}(C_{acc}^{\uparrow} - C_{acc}^{\downarrow}) \tag{10}$$

## 2.3 Extending the TEA equation to non-ideal conditions

As we discussed in Sect. (2.1), the mean advective term in Eq.(2) needs to be discarded due to the biased mean vertical wind velocity. The original formulation of the TEA method assumes a zero mean vertical wind velocity during the flux averaging



interval. This assumption is rarely valid under field conditions due to the reasons outlined earlier. Previous efforts have been focused on minimizing $\overline{w}$ to reduce the bias in the flux. However, since the wind information can not be changed after sampling, any treatments for the wind velocity measurements are final when the air samples have been collected. Thus, there is no way to guarantee a zero mean vertical velocity. A common approach to nullify mean vertical wind velocity in EC measurements is to rotate the wind coordinates in post processing to force $\overline{w}$ to zero for each averaging interval, this method - commonly referred to as double rotation - is not feasible in eddy accumulation methods. The planar fit method (Wilczak et al., 2001) is better suited for the online application in the TEA method (Siebicke and Emad, 2019). The planar fit method aligns the sonic coordinates to the long-term streamline coordinates by aligning the wind vector to the plane that minimizes the sum of squares of the vertical wind velocity means for a long period of time (weeks to months). This approach, while minimizes the vertical wind velocity means of the individual averaging intervals, does not force them to be zero. Considerable spread of $\overline{w}$ values around zero can still be observed after applying the planar fit method.

The key to extending the TEA equation to non-ideal case is to obtain an estimate of the scalar mean $\bar{c}$, and consequently remove the advective term $\overline{w}\,\bar{c}$. We achieve this by using the weighted mean of $c$ and $|w|$ and correcting for the correlation between them.

The weighted mean $(\bar{c}_{\mathrm{W}})$ of the scalar, $c$ and wind magnitude $|w|$ can be written similar to Eq. (9) as

$$\bar{c}_{\mathrm{W}} = \frac{\overline{c|w|}}{\overline{|w|}} \tag{11}$$

By decomposing into mean and fluctuating parts, we can write

$$\bar{c}_{\mathrm{W}} = \bar{c} + \frac{\overline{c'|w'|}}{\overline{w}} \tag{12}$$

It follows that

$$\bar{c} = \frac{\overline{c|w|}}{\overline{|w|}} - \frac{\overline{c'|w'|}}{\overline{|w|}} \tag{13}$$

Substituting $\bar{c}$ in Eq. (2) we can write the flux as

$$\overline{w'c'} = \overline{wc} - \frac{\overline{w}}{\overline{|w|}}\,\overline{|w|c} + \frac{\overline{w}}{\overline{|w|}}\,\overline{|w'|c'} \tag{14}$$

We can obtain all the terms in Eq. (14) from our measurements except for the covariance term $\overline{|w'|c'}$.

We define the "transport asymmetry coefficient" for the scalar $c$, $(\alpha_c)$ as the ratio of the covariance between the wind magnitude and the scalar to the covariance between the wind and the scalar.

$$\alpha_c = \frac{\overline{c'|w'|}}{\overline{c'w'}} = \frac{\rho_{c|w|}\sigma_{|w|}}{\rho_{cw}\sigma_w} \tag{15}$$





where $\rho_{c|w|}$, $\rho_{cw}$ are the correlation coefficients between $c$ and $|w|$, $c$ and $w$, respectively. $\sigma_{|w|}$ and $\sigma_w$ are the standard deviations of $|w|$ and $w$, respectively. After substitution, we write the flux as

$$\overline{w'c'} = \overline{wc} - \frac{\bar{w}}{\overline{|w|}}\,\overline{|w|c} + \frac{\bar{w}}{\overline{|w|}}\,\alpha_c\,\overline{w'c'} \tag{16}$$

Finally, we rearrange Eq. (16) and obtain the generalized TEA flux equation that gives a correct TEA flux when the mean vertical wind velocity is nonzero

$$\overline{w'c'} = \frac{\overline{wc}\,\overline{|w|} - \overline{|w|c}\,\bar{w}}{\overline{|w|} - \alpha_c\bar{w}} \tag{17}$$

### 2.3.1  Values of transport asymmetry coefficient $\alpha_c$

Calculating the correct flux using the new TEA equation (Eq. (17)) requires the knowledge of the transport asymmetry coefficient ($\alpha_c$). An analytical expression for the value of $\alpha$ can be obtained from the knowledge of the joint probability distribution of the vertical wind velocity and the scalar. If the wind and the scalar are assumed to follow a Gaussian joint probability density function in the form

$$f(w,c) = \frac{1}{2\pi\,\sigma_w\sigma_c\sqrt{1-\rho^2}} \times$$
$$\exp\left\{-\frac{1}{2(1-\rho^2)}\left(\frac{(w-\bar{w})^2}{\sigma_w{}^2} + \frac{(c-\bar{c})^2}{\sigma_c{}^2}\right.\right.$$
$$\left.\left.-2\frac{\rho\,(w-\bar{w})\,(c-\bar{c})}{\sigma_w\sigma_c}\right)\right\} \tag{18}$$

where $\rho$ is the correlation coefficient between the vertical wind velocity $w$ and the scalar concentration $c$.

We can express the analytical value of $\alpha_c$ in terms of the moments of the joint probability density function. We evaluate

$$\alpha_c = \frac{\overline{c'|w'|}}{\overline{c'w'}} = \frac{\overline{|w|\,c} - \overline{|w|}\,\bar{c}}{\overline{wc} - \bar{w}\,\bar{c}} \tag{19}$$

The term $\overline{|w|c}$ is obtained using

$$\overline{|w|c} = \int\limits_{-\infty}^{\infty}\int\limits_{-\infty}^{\infty} |w|\,c\,f(w,c)\,\mathrm{d}c\,\mathrm{d}w \tag{20}$$

Accordingly, the term $\overline{wc}$ can be obtained as follows





$$\overline{wc} = \int\limits_{-\infty}^{\infty} \int\limits_{-\infty}^{\infty} w\,c\,f(w,c)\,\mathrm{d}c\,\mathrm{d}w \tag{21}$$

After solving the integrals in Eq. (20) and Eq. (21) and substituting in Eq. (19), we find the value of $\alpha_c$ can be written using the mean vertical wind velocity and standard deviation as

$$\alpha_c = \mathrm{erf}\left(\frac{\bar{w}}{\sqrt{2}\,\sigma_w}\right) \tag{22}$$

where $\mathrm{erf}$ is the error function. Therefore, the error in the flux when when failing to account for the correlation between the scalar and the magnitude of vertical wind velocity will lead to a flux biased by the last term of Eq. (16) $\bar{w}/\overline{|w|}\alpha_c$. We can further substitute the expected value of $|w|$ by the mean of the folded normal distribution (Leone et al., 1961) and obtain an analytical expression for the expectation of the flux error due to a nonzero vertical wind velocity

$$F_{\mathrm{err}} = \overline{w}\sqrt{\pi}\left(\sqrt{2}\sigma_w \mathrm{e}^{-\frac{(\overline{w})^2}{2\,\sigma_w{}^2}} + \mathrm{erf}\left(\frac{\overline{w}}{\sqrt{2}\,\sigma_w}\right)\overline{w}\sqrt{\pi}\right)^{-1}\alpha_c \tag{23}$$

Experimental evidence has shown that different scalars behave similarly (Ohtaki, 1985; Wesely, 1988). We expect the value of $\alpha$ to be similar for different scalars due to the similar transfer mechanism. Therefore, an empirical approach for estimating $\alpha$ for one scalar is to use $\alpha$ value from another available scalar, e.g. sonic temperature.

We can express the value of $\alpha$ in terms of the updraft and downdraft contributions to the flux (flux $\uparrow$ and flux $\downarrow$) as the

difference of updraft and downdraft fluxes to the total flux.

$\alpha_c$ can be written to a good approximation as

$$\alpha_c = \frac{\mathrm{flux}^{\uparrow} - \mathrm{flux}^{\downarrow}}{\mathrm{flux}^{\uparrow} + \mathrm{flux}^{\downarrow}} \tag{24}$$

where the updraft and the downdraft contributions to the flux (flux$^{\uparrow}$, and flux$^{\downarrow}$) are defined as

$$\mathrm{flux}^{\uparrow} = \overline{c'^{\uparrow}\,w'^{\uparrow}}\,\mathrm{P}(w^{\uparrow}) \tag{25}$$

$$\mathrm{flux}^{\downarrow} = \overline{c'^{\downarrow}\,w'^{\downarrow}}\,\mathrm{P}(w^{\downarrow}) \tag{26}$$

where $\mathrm{P}(w^{\uparrow})$ is the probability of a vertical wind velocity with a positive direction, $\mathrm{P}(w^{\downarrow})$ is the probability of a vertical wind velocity with a negative direction.

The expression of $\alpha$ in terms of updraft and downdraft flux contributions is closely related to quadrant analysis. We include it here for completeness. Quadrant analysis is commonly used to inspect the contributions from different quadrants in the $(w', c')$





plane by sorting the instantaneous values into four categories according to the sign of the two fluctuating components e.g. (Katul et al., 1997; Raupach, 1981; Katsouvas et al., 2007).

We find that $\alpha$ can be written in terms of quadrants as

$$\alpha = \frac{S_1 + S_4 - (S_2 + S_3)}{S_1 + S_2 + S_3 + S_4} \tag{27}$$

Where $S_i$ is the fraction of the flux transported by contributions in quadrant $i$, given as

$$S_i = \frac{\langle\langle w'c'\rangle\rangle_i}{\overline{w'c'}} \tag{28}$$

$\langle\langle wc\rangle\rangle_i$ is the conditional average defined as

$$\langle\langle wc\rangle\rangle_i = \lim_{T\to\infty} \frac{1}{T_p} \int_0^{T_p} w'(t)c'(t)I_i \mathrm{d}t \tag{29}$$

The indicator function $I_i$ obeys

$$I_i(w',c') = \begin{cases} 1, & \text{if } (w',c') \text{ in quadrant } i \\ 0, & \text{otherwise} \end{cases} \tag{30}$$

Following the definition of Thomas and Foken (2007), the pairs $S_2$ and $S_4$ are ejections and sweeps for downward directed net flux (negative $\rho_{wc}$) and $S_1$ and $S_3$ for upward directed net flux (positive $\rho_{wc}$)

### 2.3.2   Calculating the corrected TEA flux

The new general equation for TEA (Eq. 17) extends the validity of the method to conditions where the mean vertical wind velocity is nonzero. We show here how the correct TEA flux can be calculated from the measured physical quantities.

The weighted mean over an averaging period $T_{avg}$ can be written as

$$\frac{\overline{c|w|}}{\overline{|w|}} = \overline{c|w\uparrow|}\,\frac{\overline{|w\uparrow|}T_{avg}^\uparrow}{\overline{|w|}T_{avg}} + \overline{c|w\downarrow|}\,\frac{\overline{|w\downarrow|}T_{avg}^\downarrow}{\overline{|w|}T_{avg}} \tag{31}$$

which in terms of the quantities we are measuring, translates to

$$\overline{c|w|} = \overline{|w|}\left(\frac{C_{acc}^\uparrow V^\uparrow + C_{acc}^\downarrow V^\downarrow}{V_{\text{total}}}\right) \tag{32}$$

Similarly





$$\overline{cw} = \overline{|w|} \left( \frac{C_{acc}^{\uparrow} V^{\uparrow} - C_{acc}^{\downarrow} V^{\downarrow}}{V_{\text{total}}} \right) \tag{33}$$

After substitution and simplification, we obtain the correct TEA flux in terms of the measured quantities

$$F_{\text{TEA}} = \frac{C_{acc}^{\uparrow} V^{\uparrow} \left( \overline{|w|} - \bar{w} \right) - C_{acc}^{\downarrow} V^{\downarrow} \left( \overline{|w|} + \bar{w} \right)}{\overline{|w|} - \alpha_c \bar{w}} \times \frac{\overline{|w|}}{V_{total}} \tag{34}$$

Where $F_{\text{TEA}}$ is the kinematic flux density ($\mathrm{mol\,m\,s^{-1}}$). $C_{acc}^{\uparrow}$ and $C_{acc}^{\downarrow}$ are the mean concentrations ($\mathrm{mol\,m^{-3}}$) of the scalar $c$ in updraft and downdraft reservoirs at the end of the accumulation averaging period $T_{avg}$. $V^{\uparrow}$ and $V^{\downarrow}$ are the accumulated

sample volume ($\mathrm{m^3}$) in updraft and downdraft reservoirs at the end of the averaging period. $\overline{|w|}$ is the mean of the absolute vertical wind velocity ($\mathrm{m\,s^{-1}}$) during the averaging period. $\bar{w}$ is the mean of the vertical wind velocity. $\alpha_c$ is the transport asymmetry coefficient for the scalar $c$ (dimensionless).

## 2.4   Short-time eddy accumulation

The original formulation of the true eddy accumulation method requires the samples to be accumulated for the entire averaging

interval $T_{avg}$ before the measurement can take place. This can pose limitations on the operation and the applicability of the method.

We propose a modification for the TEA method, where samples can be accumulated for a sequence of shorter intervals $\tau_i$ that add up to the averaging period $T_{avg}$. This formulation can be achieved by applying the law of total expectation to the random variable $cw$ with respect to a partitioning variable $Y$ that divides the averaging period $T_{avg}$ into multiple non-overlapping

partitions with the length $\tau_i$. It follows that the expectation of $cw$ is the conditional expected value of $cw$ given $Y$ and the flux is equal to

$$\overline{cw} = \overline{\left( \overline{(cw)|Y} \right)} = \sum_i \overline{(cw)|Y_i} \, \mathrm{P}(Y_i) \tag{35}$$

This allows to write Eq. (8) as a sum of $j$ intervals with the length of $\tau_i$ and a scaling factor $A_i$ each. The concentration for either updraft or downdraft reservoirs can then be calculated from

$$C^{\uparrow\downarrow} = \frac{1}{\sum_{i=1}^{i=j} A_i \overline{|w_i|} \tau_i} \sum_{i=1}^{i=j} \frac{1}{\int_t^{t+\tau_i} A_i |w|} \int\limits_t^{t+\tau_i} A_i |w| \, c \, dt \tag{36}$$

The concentration in either updraft or downdraft reservoirs at the end of the averaging interval is the mean of the short interval concentration measurements $C_i$ weighted by the sample volume during the short interval $V_i$.





$$C^{\uparrow\downarrow} = \frac{1}{V_{total}} \sum_{i=1}^{i=j} C_i^{\uparrow\downarrow} \, V_i \tag{37}$$

We call this new variety of eddy accumulation, the short-time eddy accumulation method (STEA).

## 2.5 Effect of buffer volumes

The short-time eddy accumulation method can be achieved in two ways, either using expandable buffer volumes (e.g. bags), which are emptied after each short interval measurement $C_i$ or using a flow-through system with rigid buffer volumes. The flow-through system has a practical operational benefit but requires additional correction to reverse the effect of buffer volumes on the concentration signal. Buffer volumes act as low pass filters (Cescatti et al., 2016). They attenuate the magnitude of the

high-frequency part and shift the phase of the signal. The buffer concentration at time step $n$ is dependent on the new input sample concentration and the buffer concentration from the previous step $y[n-1]$. Thus, the buffer volume concentration $y_n$ response to an input $C_i$ can be described with the difference equation

$$y_{[n]} = C_{i[n]} \, \dot{q}_i + (1 - \dot{q}_i) \, y_{[n-1]} \tag{38}$$

where $\dot{q}$ is a dimensionless flow rate that is defined as the ratio between the sample mass to the total mass of air in the buffer

volume, at each time step $n$

$$\dot{q_n} = \frac{V_i \, \rho_i}{V_b \, \rho_b} \tag{39}$$

Where $V_i$ and $\rho_i$ are the volume and density of the short accumulation sample, respectively, while $V_b$ and $\rho_b$ are the volume and the air density of the buffer, respectively. Equation (38) characterizes a first-order linear filter. The treatment as a discrete-time process aligns with the discrete operation of the STEA method.

The system response is characterized by the dimensionless flow rate. The time constant of the system is defined as the required time for the system to reach $1/e$ from a step increase and relates to $\dot{q}$ by (Taylor et al., 2013).

$$\tau = -\frac{\Delta t}{\ln(1 - \dot{q})} \tag{40}$$

where $\Delta t$ is the sampling interval.

## 2.6 Methods

### 2.6.1 Experimental site

Flux measurements were performed over a flat agricultural field of the Thünen Institute, located at 52.297 N, 10.449 E in Braunschweig, Germany. The site has an altitude of 76 m above sea level.





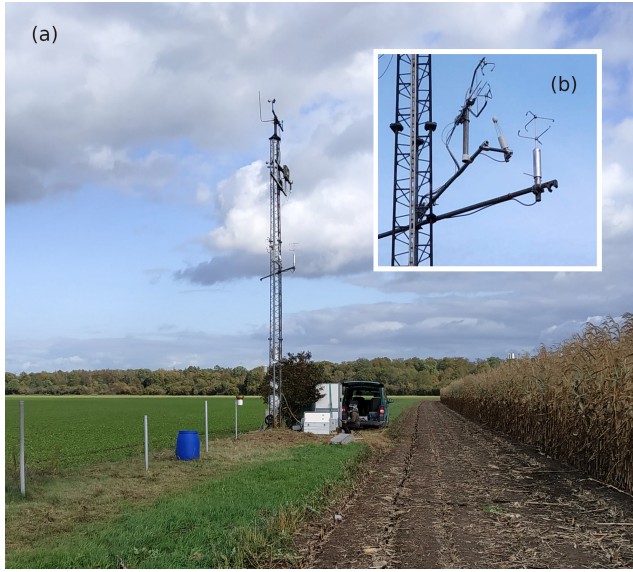

**Figure 1.** Photograph of the experimental field site showing the measurement tower **(a)** and a close up on the flux instruments mounted on the tower **(b)**.

### 2.6.2 Experiment period

The experiment started in September 2019. The first period (September 2019 to April 2020) was used for the development of the method which coincided with winter. The measurements during this period showed a lower quality and were excluded from the analysis.

The winter period was followed by a period of stable operation, running from May 2020 until October 2020. During this period, we had a stable and continuous operation, however, intermittent discontinuities still existed due to blocked inlets, rain, or technical failures. From this period, we selected six weeks in summer, spanning from 18 June 2020 to 31 July 2020, to compare the different methods.

### 2.6.3 Instruments

EC and STEA measurement complexes were mounted at 5 m height above the ground (Fig. 1). The instruments used in the experiment for flux measurements and data analysis are listed in Table 1. Meteorological variables were logged using a Sutron 9210 XLite logger (Sterling, USA). All the raw data needed for flux processing were synchronized on the STEA computer and remote servers for real-time processing.

The EC system comprised a dedicated sonic anemometer (uSonic-3 Omni H) and an open-path infra-red gas analyzer (IRGA). Wind and scalar density data were acquired at 20 Hz frequency. Relative to the Class-A sonic anemometer used for STEA, the northward, eastward, and vertical separation of the IRGA was $-17$ cm, 26 cm, and $-15$ cm, respectively. The





**Table 1.** Variables and instruments. Manufacturer key: METEK GmbH (Elmshorn, Germany), LI-COR Environmental Inc. (Lincoln, Nebraska, USA), LGR, (Los Gatos Research Inc., USA), Bosch (Bosch Sensortec GmbH, Germany), Vaisala (Helsinki, Finland), Kipp & Zonen (Delft - The Netherlands), Delta-T Devices Ltd (UK), Stevens Water Monitoring Systems, Inc (Oregon, USA), Texas Electronics (Dallas, USA)

| Variable | Sensor | Manuf. | Method | Freq. |
|---|---|---|---|---|
| Wind $u, v, w$ | uSonic-3 Omni H | METEK | EC | 20 Hz |
| Sonic temp. Ts | uSonic-3 Omni H | METEK | EC | 20 Hz |
| Wind $u, v, w$ | uSonic-3 Class A | METEK | TEA | 10 Hz |
| Sonic temp. Ts | uSonic-3 Class A | METEK | TEA | 10 Hz |
| $CO_2$ density | LI-7500A | LI-COR | EC | 10 Hz |
| $H_2O$ density | LI-7500A | LI-COR | EC | 10 Hz |
| $CO_2$ ppm | FGGA-24r-EP | LGR | TEA | 1 Hz |
| $H_2O$ ppm | FGGA-24r-EP | LGR | TEA | 1 Hz |
| $CH_4$ ppm | FGGA-24r-EP | LGR | TEA | 1 Hz |
| Air pressure P | BME280 | Bosch | TEA | 50 Hz |
| Air temperature | BME280 | Bosch | TEA | 50 Hz |
| Air humidity | HMP155 | Vaisala | Meteo | 10min |
| Air temperature | HMP155 | Vaisala | Meteo | 10min |
| Net radiation | CNR4 | KIPP | Meteo | 10min |
| Global radiation | BF5 | DELTA-T | Meteo | 10min |
| Soil heat flux | HFP01 | LI-COR | Meteo | 10min |
| Soil moisture | SDI-12 | Stevens | Meteo | 10min |
| Precipitation | TR-525M | Texas Elec. | Meteo | 10min |

Class-A sonic had a north offset azimuth of $90°$ degrees. Relative to the Omni-sonic anemometer, the northward, eastward, and vertical separation of the IRGA was $20\,\text{cm}$, $-15.3\,\text{cm}$, and $-20\,\text{cm}$. The north offset of the Omni-sonic was $169°$ degrees.

## 2.7  TEA system description

The TEA system used in the experiment is based on an earlier system of Siebicke and Emad (2019). The new system used the same mass flow controllers and shared most of the operating software. It has, however, several differences and improvements. One major difference is the use of fixed stainless steel buffer volumes instead of expandable bags. The system was developed initially as a hybrid TEA-EC method to run the TEA method in a flow-through mode (Siebicke, 2016). The system was set up to operate in the STEA flow-through mode described earlier in the theory section. A constant duration for the short intervals ($\tau_i$) equal to one minute was used. The STEA system is comprised of two identical sampling lines, one for updrafts and one for downdrafts. Each of the sampling lines has two rigid buffers in a sequence connected using $6\,\text{mm}$ Teflon tube (Fig. 2).

The STEA sampling inlets were installed in close proximity to the sonic's center of measurement volume. The horizontal separation was $22\,\text{cm}$, while the vertical separation between the two inlets was $2\,\text{cm}$. Upon sampling, the collected samples were carried using $6\,\text{mm}$ Teflon tubes to the first set of buffers. The sampling can be summarized in the following steps (see a detailed description of the system operation and sampling in (Siebicke and Emad, 2019)):





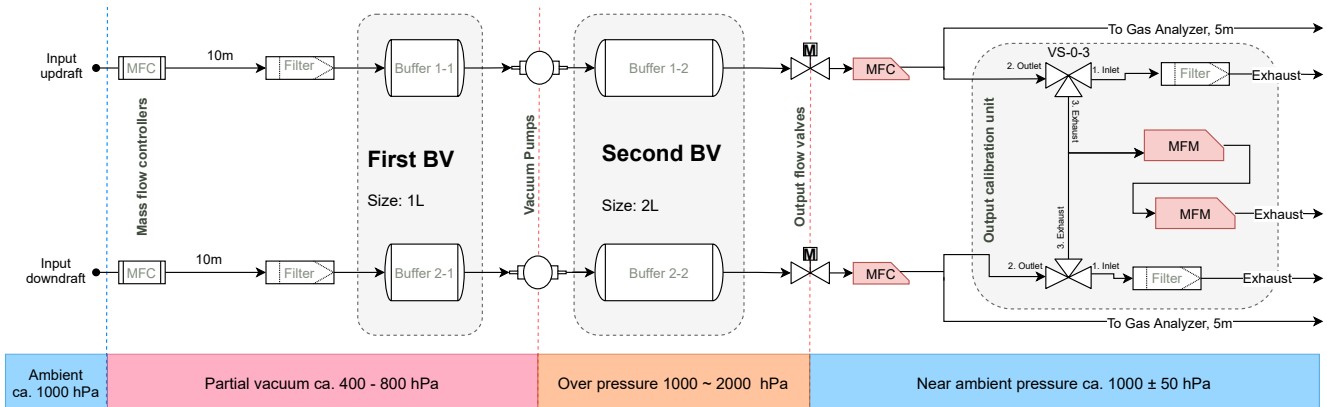

**Figure 2.** Functional and hydraulic schematic of the implemented flow-through STEA system showing components, layout, properties, and operation conditions. Air samples are collected at the input and travel in distinct sampling lines for updrafts and downdrafts. Samples travel through tubes (lengths are shown), through filters, are then collected into two sets of buffer volumes shown here as *First BV* and *Second BV* separated by two vacuum pumps. The "Output flow valves" followed by mass flow controllers, control the output flow rate from the second set of buffers to the gas analyzers. Finally, samples can optionally be forwarded to a set of mass flow meters used for calibration purposes. The colored bottom bar below shows the range of pressure values at each stage.

1. 3D wind measurements are acquired from the sonic anemometer (uSonic-3 Class A) with a 10 Hz sampling frequency.

2. Wind coordinates are rotated into the streamline coordinates using the planar fit method without an intercept (Dijk et al., 2004). The fit is performed online as a running window operation with a window width of 2 days and an update frequency of once every 30 minutes.

3. The mean vertical wind from the previous window width (30 min.) is removed to minimize $\overline{w}$. This is equivalent to applying a high-pass filter to the vertical wind velocity measurements.

4. Sampling line is determined (updraft or downdraft) based on the direction of the rotated vertical wind velocity component.

5. Sampling scaling factor $A_i$ is calculated based on wind conditions in the near past and the calibration coefficients of the mass flow controllers. The scaling factor should be constant during the short accumulation intervals.

6. Air samples are collected, the controllers are adjusted to collect an air sample with a volume equal to $A_i |w|$.

7. When enough sample volume is accumulated in the respective buffer volume, samples are forwarded to the gas analyzer for analysis. The amount of sample volume needed is determined based on the required flow rate for the gas analyzer and the time needed to flush the tubes and the measurement cell and to perform enough repeated measurements.

8. Mean concentrations of accumulated samples are measured. The slow gas analyzer (LGR FGGA-24r-EP) alternates on measuring the concentrations $C_i$ of the accumulated samples for updraft and downdraft. The accumulation time for



the short intervals was set to a fixed interval of one minute instead of an adaptive interval duration. During each short

interval, the gas analyzer performs repeated measurements for the gas concentration.

## 2.8  STEA flux computations

This section describes the steps followed to obtain the final and corrected STEA flux. Firstly, we discuss the effect of water vapor on the measured concentrations of other scalars and how we corrected that remaining water cross-sensitivity. Then, we present the procedure of data quality screening. Next, we detail the steps of calculating the final STEA flux. Finally, we present

the buffer volume empirical correction we applied.

### 2.8.1  Water correction

The gas analyzer used for the STEA measurements (LGR FGGA-24r-EP) reports the molar fraction of $CO_2$ and $CH_4$ of moist air in parts per million (ppm). The measurements of $CO_2$ can not be used directly, as they are affected by the presence of water vapor. The presence of varying water vapor concentrations in the sample affects the measurements of $CO_2$ and $CH_4$ in cavity

ring-down spectroscopy instruments in at least two ways: (i) the dilution effect, and (ii) the spectroscopic line broadening (Rella, 2010). Rella (2010) proposed a quadratic equation to correct for the combined effect of line broadening and water vapor dilution. The correction involves estimating a the parameters ($a$) and ($b$) in the equation

$$r_c = \frac{\chi_c}{1 + a\chi_w + b\chi_w^2} \tag{41}$$

where $r_c$ is the dry mole fraction of the species $c$, $\chi_c$ is the wet mole fraction measured by the instrument, and $\chi_w$ is

the water mole fraction measured by the instrument. For the LGR gas analyzer, these coefficients were estimated as $a = -1.219 \times 10^{-06}(\pm 2.169 \times 10^{-09})$, $b = 1.229 \times 10^{-12}(\pm 1.073 \times 10^{-13})$ (Hiller et al., 2012) for $CO_2$ where the unit is ppm.

We found that using the same parameters could not control for all the effects of water. A linear slope different from zero was still found when supplying the gas analyzer with air of varying water signal and constant $CO_2$. This suggests a remaining cross-sensitivity on the presence water vapor. To control for this small cross-sensitivity we used a linear fit to obtain the slope

and corrected for it.

We were not able to supply the gas analyzer with air of known $CO_2$ signal. Instead, we used the systems buffer volume to collect air from the atmosphere, closed the inlets, and supplied the gas analyzer with enough sample flow rate for measurement. The accumulated sample was enough to supply the gas analyzer for ca. 10 min. We repeated the measurements several times and used the obtained dataset for correcting the renaming cross-sensitivity.

### 2.8.2  Raw data quality screening

Raw data were processed to ensure the removal of outliers due to measurement errors and instruments malfunction. This included the following steps





– Despiking: gas analyzer and wind measurements are screened for outliers and removed following a procedure similar to Vickers and Mahrt (1997).

– Dropouts removal: some sensors would get stuck on one value, the first value is kept and the rest are discarded.

– Plausibility limits: values falling outside physical ranges were removed. Limits used are similar to those of Sabbatini et al. (2018).

– Deadband removal: measurement of short interval events involve regularly switching the sampling line coming to the gas analyzer from updraft to downdraft reservoirs. This will cause contamination from subsequent samples. We experimentally chose a 25-second deadband at the start of each short interval event. The measurements falling within the deadband were removed. Figure 3 shows an example of deadband removal at the start of each averaging interval.

– Detection of sample contamination: periods where the flow rate to the gas analyzer is smaller than $400 \mathrm{~mL} \mathrm{~min}^{-1}$ are flagged. Under these conditions ambient air might enter the system and contaminate the collected samples. When the number of flagged data points exceeds $10\%$ of the total points in the sampling interval, data in the sampling interval are discarded.

### 2.8.3 STEA flux calculation

After measurements are quality checked and erroneous data points are excluded, the final STEA flux is calculated as follows

– Short interval statistics: for each short interval sample, the gas analyzer will have several repeated measurements for the concentrations $C_i$, however, only one value is needed for the flux calculation. We use the median to obtain the representative value in order to minimize uncertainty and exclude outliers. Figure 3 shows an example of data quality checking and choice.

– Calculate air molar volume: the molar volume of air is needed to express the flux in dynamic flux units. The molar volume is calculated using sonic temperature, pressure, and humidity measurements.

– Calculate short intervals weights: following Eq. (37), the measured short interval concentration should be weighted by the ratio of the accumulated volume during that interval to the total buffer volume $V_i/V_{tot}$.

– Calculate values of $\alpha_\theta$: values of $\alpha_\theta$ are calculated using vertical wind velocity and sonic temperature measurements. Values of $\alpha_\theta$ larger than 1 are discarded as they indicate a problem with the measurement.

– Calculate updraft and downdraft mean concentrations: $C_{acc}^{\uparrow}$ and $C_{acc}^{\uparrow}$ are calculated for the averaging period $T_{avg}$.

– Calculate the flux: the STEA flux equation shown in Eq. (31) is used to obtain the final flux.





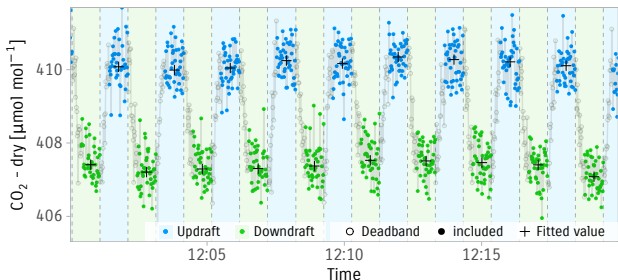

**Figure 3.** Data choice and fitting procedure. Points represent consecutive concentration measurements from the gas analyzer. Updraft and downdraft samples are highlighted with blue and green, respectively. Grey hollow points are excluded from the data fitting (deadband). Cross points are the chosen representative concentrations for each short interval. Data are from 21 June 2020 at mid day.

### 2.8.4 Buffer volume empirical correction

The use of buffer volumes introduces systematic biases to the fluxes. Buffer volumes act on the signal as a low pass filter. The time constant for the filter is needed to estimate the magnitude of filtering. We used Eq. (40) to estimate the time constant of the buffers. For each of the buffer volumes, a measurement point is acquired every two minutes. The mean dimensionless mass flow rate to the gas analyzer is estimated from the pressure, the volume, and the estimated volumetric flow rate to the gas analyzer. The average time constant was estimated to be 18 minutes.

We simulated the effect of buffer volumes on the high-frequency sonic temperature signal. The loss in the fluxes was parameterized by artificially degrading the sonic temperature in a procedure similar to Goulden et al. (1996) and Berger et al. (2001)

### 2.9 EC reference flux measurements and computations

The raw data from the two sonics and the high-frequency gas density measurements from the IRGA were used to compute eddy covariance fluxes for water vapor and $CO_2$ in the period from 1 April 2020 to 1 November 2020 using EddyPro® software (LI-COR Env. Inc. USA) version 7.0.4. The flux processing steps were chosen to be as similar as possible to the TEA processing scheme. The calculation involved the following steps:

– Statistical screening for the data quality issues following (Vickers and Mahrt, 1997), including tests for spikes, amplitude resolution, dropouts, absolute limits, and higher moment statistics.

– De-trending of raw time series by block averaging.

– Compensation of the time lag between the wind and the scalar time series using covariance maximization.

– Tilt correction using the planar fit method without an intercept (Dijk et al., 2004). The planar fit procedure was performed in two ways. First, for the entire experiment period, and second, as a running window operation with a width of 2 days, similar to the procedure performed online by the TEA system.





– Analytical high and low-frequency corrections to correct for the spectral attenuation of the IRGA (Moncrieff et al., 2005, 1997)

## 2.10 Data selection for method comparison

For comparing the fluxes calculated from both methods, we selected averaging intervals according to the following criteria:

– Spike removal: following Vickers and Mahrt (1997) using a window width of 6 hours and a threshold of 2 standard deviations.

– Rainy periods exclusion: data records where rain was recorded were excluded.

– Flux quality flags: periods where the flux quality flag is 1 or 2 according to Foken et al. (2005) were excluded.

– STEA low flow rate: averaging intervals flagged with the low flow rate flag described earlier were discarded.

After applying the above criteria, 1971 averaging intervals remained. They accounted for $54.1\%$ of the whole comparison period. Table 2 shows a summary of data quality checks results.

**Table 2.** Summary of data quality checks for STEA and EC fluxes used in the EC/STEA flux intercomparison showing for each criterion, the number of averaging intervals that were excluded and the ratio of the excluded averaging intervals to the total. Details on the criteria and the thresholds used are provided in Sect. 3.4

| Criteria | Averaging intervals | Ratio (%) |
|---|---|---|
| EC missing value | 25 | 0.7 |
| Spikes | 41 | 1.1 |
| Technical failure | 75 | 2.1 |
| Rain | 182 | 5.0 |
| STEA low flow rate | 214 | 5.9 |
| Flux quality flag 2 | 390 | 10.7 |
| Flux quality flag 1 | 743 | 20.4 |
| OK data | 1971 | 54.1 |

To compare the overall difference between the two methods, we used the coefficient of determination $R^2$ and the slope of the orthogonal distance regression (ODR) (also known as major-axis regression and model II regression). ODR considers the errors in $x$ and $y$ as opposed to OLS regression which assumes the error in $x$ is negligible (Wehr and Saleska, 2017).

## 2.11 Numerical simulations

In addition to the experiment described earlier, we set up a numerical simulation to test the magnitude of the error due to nonzero $\bar{w}$ on the flux. We used high-frequency measurements obtained from the IRGA and the sonic anemometer during one





week from 19 June 2020 to 26 June 2020. We added a random $\bar{w}$ offset in the range ($-0.25$ to $0.25\,\mathrm{m\,s^{-1}}$) to each averaging

interval. We obtained 6 repetitions and calculated the flux according to different formulas. In total, there were about 1800

30-minute averaging intervals. The methods compared were: i) the flux calculated using the concentrations formula of Hicks

and McMillen (1984) shown in Eq. (10), ii) the equation for DEA including the non equal volume correction of Turnipseed

et al. (2009), and iii) the new generalized equation proposed in the current study (Eq. 16) utilizing $\alpha_\theta$ values calculated from

sonic temperature.

We applied minimal quality checks on the resulting fluxes before the comparison. We applied the steady state test following

Foken et al. (2005), which removed $24\%$ of the averaging intervals. We limited the values of $|\alpha_\theta|$ to less than 0.7, which

removed an additional $11\%$ of the averaging intervals. The excluded averaging intervals occurred almost exclusively during

low developed turbulence and night-time conditions.

## 3   Results and Discussion

We first discuss the problem of nonzero mean vertical wind velocity and the new generalized TEA equation. Then, we discuss

the value and interpretation of the transport asymmetry coefficient $\alpha$. Next, we discuss the newly proposed short-time eddy

accumulation method. Then, we discuss some results and aspects of the STEA flux calculations. Afterwards, we will present

the flux intercomparison between STEA and EC. Finally, we discuss the effect of using fixed buffer volumes on the fluxes and

the proposed empirical correction.

### 3.1   Nonzero mean vertical wind velocity

We presented a new formulation for the TEA equation. The new equation (Eq. (16)) employs information about the scalar

transport to allow the estimation of $\bar{c}$ from available TEA measurements and, consequently, constraining the bias term $\bar{w}\bar{c}$.

Besides the correction of the nonzero $\overline{w}$ bias, the estimation of the scalar mean $\bar{c}$ is essential for the WPL correction and the

calculation of storage fluxes.

The terms of Eq. 16 account for different contributions to the flux. The first term on the right hand side is equivalent

to calculating the flux as the difference in accumulated mass between updraft and downdraft. When $\bar{w} = 0$, the equation is

reduced to this term only. The second term accounts for the bias introduced by the advective term $\bar{w}\bar{c}$ by using the weighted

mean of the scalar and the magnitude of wind as an estimate for $\bar{c}$. We show that when $\bar{w} \neq 0$, the first two terms are equivalent

to using the concentration formula of Hicks and McMillen (1984) shown in Eq. (10), with the unequal volume correction

of Turnipseed et al. (2009) that accounts for the small difference between the weighted mean $\bar{c}_W$ and average of concentrations

$(C_{acc}^\uparrow + C_{acc}^\downarrow)/2$. Refer to appendix A for details about this equality. The new third term $\overline{c'|w'|}/\overline{|w|}$ corrects for the correlation

between the scalar and the magnitude of the wind. Ignoring the third term will result in a flux biased with the ratio $\bar{w}/\overline{|w|}\alpha_c$.

The new TEA equation reveals an important insight. When using the new equation to calculate the flux, the error in the flux

when $\overline{w} \neq 0$ is independent of the scalar concentration and is governed by the characteristics of the turbulent transport. This





gives higher confidence in using the TEA method for measuring atmospheric constituents with high background concentration

and small flux (low deposition velocity).

To quantify the effect of nonzero $\overline{w}$ on the fluxes, we used the results of the numerical simulation explained in the methods section. The results of the comparison are presented in Fig. 4. Additionally, we used the slope and the coefficient of determination, $R^2$ obtained from a linear fit of the calculated fluxes against the reference EC flux and the mean absolute difference, MAD to compare the different formulations.

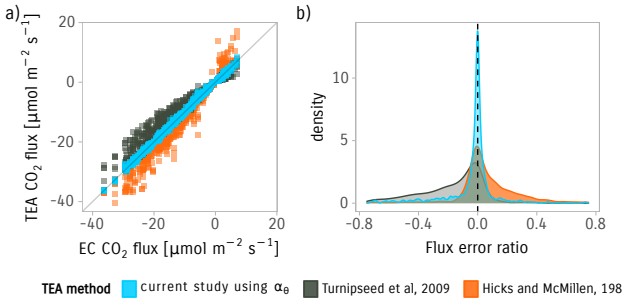

**Figure 4.** a) Comparison of different equations of TEA flux calculation against a reference EC flux. Data were obtained from a numerical simulation using high frequency measurements over a one-week period from 19 June 2020 to 26 June 2020. We added a random $\overline{w}$ offset in the range (-0.25 to 0.25 $\mathrm{m\,s^{-1}}$)). Colors represent different formulas. b) Kernel density estimates of the flux error ratio using the three different formulas.

We found that by ignoring the correction completely and using the concentration TEA equation of Hicks and McMillen (1984), the TEA flux was overestimated. The slope of the linear fit was $1.09$ ($R^2 = 0.96$) and the mean absolute difference (MAD) was $0.97$ $\mathrm{\mu mol\,m^{-2}\,s^{-1}}$. Using the DEA equation of Turnipseed et al. (2009) which includes an additional term to correct for the effect of unequal volume on the flux, the linear fit slope was $0.88$ ($R^2 = 0.95$) and the mean absolute difference was $\mathrm{MAD} = 1.03$ $\mathrm{\mu mol\,m^{-2}\,s^{-1}}$. The correction of nonzero $\overline{w}$ using $\alpha_\theta$ significantly improved the match. The linear fit

produced a slope of almost 1 ($R^2 = 0.997$). The mean absolute difference was $0.26$ $\mathrm{\mu mol\,m^{-2}\,s^{-1}}$ almost four times smaller than the other two methods. These results indicate that the proposed correction significantly reduces the bias and uncertainty of TEA fluxes when $\overline{w} \neq 0$.

### 3.1.1   Value and interpretation of the transport asymmetry coefficient $\alpha$

We proposed the transport asymmetry coefficient $\alpha_c$, defined in Eq. (15) as the ratio between the covariance of the scalar

and the absolute value of vertical wind $\overline{|w'|c'}$ to the covariance between the scalar and the vertical wind $\overline{w'c'}$. The value of $\alpha_c$ indicates the disparity of the flux transport between updrafts and downdrafts. We showed the analytical expectation of $\alpha$ based on the assumption of a joint Gaussian distribution in Eq. (19). This assumption, although used in the literature e.g. (Wyngaard and Moeng, 1992), is not adequate. While the wind might be normally distributed for most stability classes (Chu et al., 1996), the scalar can depart significantly from normality (Berg and Stull, 2004). Other distributions might be more





suited for approximating the joint probability distribution (Frenkiel and Klebanoff, 1973). For example, Katsouvas et al. (2007) showed using experimental data that a third-order Gram–Charlier distribution was necessary and sufficient in most of the cases for describing the quadrant time and flux contributions.

We found using high-frequency measurements that the value of $\alpha$ for $CO_2$ correlates moderately with the skewness of the measured scalar ($r = 0.61$). On average, updrafts have larger contribution to the flux. The mean of $\alpha$ for $CO_2$ and sonic

temperature calculated from high-frequency measurements for periods with negligible $\bar{w}$ was approximately $0.2$ under unstable and good turbulent mixing conditions ($|\rho_{wc}| > 0.25$) with a standard error, $SE = 0.01$. Under stable stratification ($\zeta > 0$), the mean of $\alpha$ was approximately equal to $-0.18$ but with a higher spread around the mean, $SE = 0.09$. These values generally agree with values found from studies using conditional sampling (Greenhut and Khalsa, 1982) and LES simulations (Wyngaard and Moeng, 1992) which found that updrafts contribution to the flux is 2 to 3 times larger than downdrafts.

We showed that the bias in TEA flux when $\bar{w} \neq 0$ is dependent on the value of $\alpha_c$. For TEA, $\alpha_c$ is not readily available, since its calculation requires the high-frequency information of the scalar. Similarity of scalar transport suggests that $\alpha$ values for different scalars should be similar, allowing the use of $\alpha_\theta$ calculated from sonic temperature as a substitute for $\alpha_c$. The similarity was confirmed empirically by calculating the values of $\alpha_\theta$ and $\alpha_c$ from high-frequency measurements. A linear fit with a slope of $0.98$ and $R^2$ of $0.962$ was obtained during steady-state and well-developed turbulence conditions. During such conditions,

$\alpha_\theta$ can substitute $\alpha_c$ to calculate the flux correction ratio. However, the correction becomes large and unreliable in periods where $\sigma_w$ and $\rho_{cw}$ are small, associated with small fluxes during night-time and stable conditions. Additionally, temperature is considered a bad proxy during near-neutral conditions (McBean, 1973; Hicks et al., 1980) due to its contribution to buoyancy. We noticed that the variance in $\alpha$ values is higher under weakly developed turbulence. We experimentally determined the threshold for the safe use of $\alpha$ for correction as $|\rho_{cw}| = 0.2$. Below this threshold, values of $\alpha$ larger than 0.5 are observed,

making the correction unreliable. This threshold can be seen as an indicator for the violation of assumptions of homogeneity and stationarity or other problematic conditions. Similar uses for the correlation coefficient are common in the literature e.g. (Foken and Wichura, 1996).

We determined experimentally that the error in the flux due to nonzero $\bar{w}$ becomes significant (larger than $10\%$ of the flux) when $\bar{w}$ exceeds $0.21\sigma_w$ for periods with good turbulent mixing conditions ($|\rho_{w,CO_2}| > 0.2$). This threshold is close to

the analytical value of $0.323\,\sigma_w$ obtained from the Gaussian joint probability distribution. To push this threshold further, $\alpha_\theta$ calculated from sonic temperature can be used during good turbulent mixing conditions ($|\rho_{w,CO_2}| > 0.2$). Simulations indicate that the average relative confidence interval for the predicted value of $\alpha_\theta$ from $\alpha_{CO_2}$ is $0.17\%$ of the fit value. In summary, to keep the error in the flux below $10\%$, $\alpha_\theta$ can be safely used to correct for biased $\bar{w}$ as long as $\bar{w} < 0.7\sigma_w$. This limit is considered forgiving and easy to achieve with online coordinates rotation and other simple online treatments. The only times

where this limit is expected to be reached is when $\sigma_w$ is very small (e.g. during night-time conditions) where other problems such as low turbulence mixing and violations of the method's assumptions are expected to occur. These periods largely overlap with periods considered of low quality and are usually excluded from the analysis (Foken et al., 2012b).



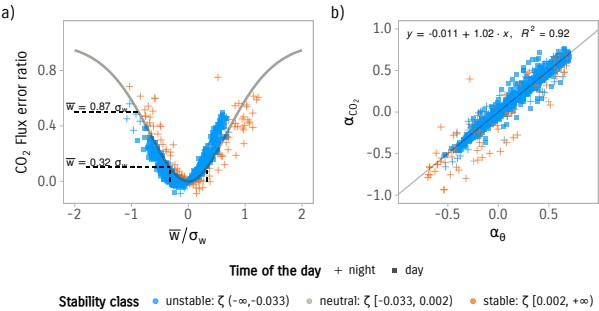

**Figure 5.** a) The error ratio in the $CO_2$ flux calculated using the TEA method due to nonzero mean vertical wind. The solid gray line represents the analytical values of the error in the flux (if a joint Gaussian probability distribution is assumed). The points are the observed error calculated from high-frequency measurements colored according to stability classes. The point shape distinguishes day-time and night-time data. b) Relation of $\alpha_{CO_2}$ calculated for $CO_2$ and and $\alpha_\theta$ calculated from sonic temperature and a 1-to-1 line for reference.

## 3.2 Short-time Eddy Accumulation

We proposed the short-time eddy accumulation method (STEA), a modification of the TEA method where the accumulation
time is divided into shorter intervals of time that add up to the flux averaging interval. The accumulation on shorter time scales brings many advantages. First, it allows adapting to the local range of vertical wind velocity values which improves the resolution and dynamic range of the system. This can be achieved by exploiting the autocorrelation of the wind velocity signal to predict a scaling parameter $A_i$ better adapted to the local velocity field for each interval. For a short interval, the range that the sampling apparatus need to cover will be on average smaller than the range of the whole averaging interval. We found for
a short averaging interval of one minute, the range was on average $60\%$ smaller than that of the whole flux averaging interval. As a result, the upper bound of the required dynamic range for $w$ reported by Hicks and McMillen (1984) as $5\,\sigma_w$ is lowered to $3.33\,\sigma_w$.

Additionally, the accumulation on varying intervals means the measurement frequency can be adjusted to match that of the gas analyzer or the precision requirements. This can be useful for reactive species and other traces gases, where relatively fast
gas analyzers are available but not fast enough for EC.

Figure 6 demonstrates how the method works. In this example, the high-frequency samples are collected at 5 Hz frequency for a 30-minute long averaging interval. The averaging interval is divided into 30 short intervals with a duration varying from 70 to 190 seconds. The flux in this example equals $-14.24\,\mu\mathrm{mol\,m^{-2}\,s^{-1}}$.

Finally, the STEA method facilitates using the STEA system in flow-through mode using rigid reservoirs. The operation in
flow-through mode requires two sets of buffer volumes in a series. The ideal operation of such a system can be achieved as follows:





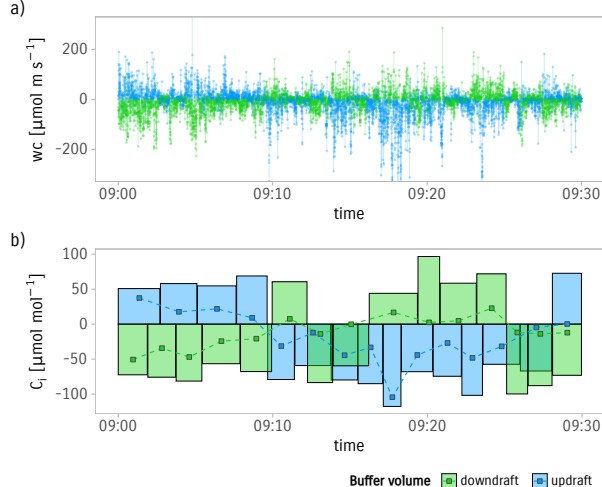

**Figure 6.** Sample accumulation using the STEA method. An example of 30 minutes of measurements: A) samples $wc$ are collected based on wind direction and proportional to its magnitude; B) Short intervals are accumulated, the variable short interval duration guarantees equal accumulated volume for consecutive short intervals. Points are the concentrations $C_i$ measured by the gas analyzer. The area of each rectangle represents the accumulated sample volume in arbitrary units and is equal to the relative weight for each concentration measurement. The sum of all measurements $C_i$ weighted by the relative sample volume will equal the covariance. Data are from 20 June 2020.

1. Wind is measured, rotated, and the value of the scaling parameter $A_i$ is updated based on wind statistics and calibration parameters.

2. Air samples are collected into the first set of buffer volumes according to vertical wind sign and proportional to the vertical velocity magnitude and the value of $A_i$ until a predefined accumulated volume is reached.

3. When the goal accumulated volume is reached, the second set of buffers is disconnected from the first. Sample accumulation time $\tau_i$ and accumulated mass are recorded. Samples are forwarded to the gas analyzer.

4. The slow gas analyzer alternates on measuring scalar concentration for each interval $C_i$ from the second set of buffers for updraft and downdraft.

It is important for this scheme to keep the mass flow rate to the second set of buffers constant so that the assumption of time invariance of the linear filter used to model the buffers is not violated.

### 3.3 STEA fluxes computations

In this section, we will discuss some aspects related to the calculation of the STEA fluxes. We first discuss the effects of water vapor on $CO_2$ concentration measurements. Then, we discuss the effect of coordinates rotation on the fluxes. Finally, we discuss the effect of density fluctuations on eddy accumulation methods.





### 3.3.1 Water correction

Treatment of the residual cross-sensitivity of $CO_2$ on water signal using a linear fit produced a small slope of $-1.17 \times 10^{-4}$ shown in Fig. (7). Thus, a difference in water concentration of $4000\,\mathrm{ppm}$ between updraft and downdraft reservoirs, typically observed at extreme conditions, will lead to a difference on the order of $0.5\,\mathrm{ppm}$ for $CO_2$.

Applying the water correction using the quadratic fit and the slope correction reduced the STEA fluxes in comparison to the direct calculation of mixing ratios. However, it improved the fit between the STEA and the reference EC flux ($R^2$ increased from $0.81$ to $0.85$.)

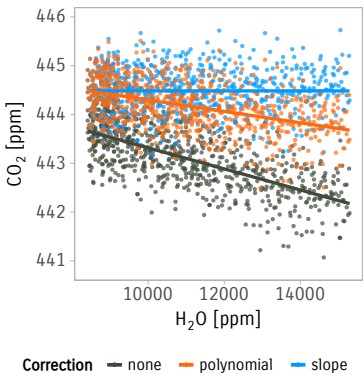

**Figure 7.** Effect of water correction on the measured $CO_2$ concentration using the LGR FGGA-24r-EP instrument. Points represent measured $CO_2$ by the gas analyzer when air with constant $CO_2$ concentration and varying $H_2O$ concentration was supplied. Lines represent linear regression fits. Red colored points and line represent $CO_2$ measurements after applying the polynomial correction (Hiller et al., 2012; Rella, 2010). In blue are the $CO_2$ measurements after applying our slope adjustment correction to remove additional cross-sensitivity on water.

### 3.3.2 Coordinates rotation

The online coordinates rotation produced stable rotation angles over the experiment period. The eddy covariance fluxes calcu-
lated using the Class-A sonic using a two-month long dataset (1 June 2020 to 1 August 2020) produced the rotation angles: x-Pitch $= 0.6°$; y-Roll $= -4.3°$ (using the YXZ Euler convention). Whereas for the TEA moving-window online rotation, larger pitch angles were observed with a mean of $3.6°$ and values slowly climbing from $1.2$ to $6°$ during the 6 weeks comparison period. The roll angle ranged from $-0.9°$ to $-0.24°$ with an average of $-0.4°$.

    The use of online rotation with a moving window of two days minimized the residual vertical mean wind in comparison to
using the whole period of the experiment. This is likely due to a better adaptation to the local wind field. The mean vertical wind values ranged from $-0.05\,\sigma_w$ to $0.38\,\sigma_w$ (mean magnitude of $0.251\,\sigma_w$) compared to $-0.04\,\sigma_w$ to $0.04\,\sigma_w$ (mean magnitude of $0.06\,\sigma_w$) when using the online rotation with a short moving window.





To estimate the effect of the online rotation method on the fluxes, we calculated EC fluxes using the two different rotation approaches while keeping other treatments constant. The comparison revealed that the online rotation with a moving window had minimal effect on the fluxes: a slope of 1 and $R^2$ of 0.98 were obtained when using a linear fit. Nevertheless, this comparison only included data of good quality. A comprehensive comparison might be needed to identify the effect under non-ideal conditions.

### 3.3.3 Effect of density fluctuations

Changes in air density due to temperature, pressure, and dilution of water vapor and other gases bias the measured flux. If the density of a scalar is measured, a correction is needed to account for these effects (Webb et al., 1980). However, If the mixing ratio is measured, no correction is required since it is a conserved quantity. In TEA and STEA, after samples are collected and mixed in buffer volumes, the mean mixing ratio is measured. Therefore, no correction for density effects is needed. The resulting flux is equivalent to the flux measured with mixing ratios $\overline{r'_c w'}$. However, the density fluctuations might affect TEA and STEA differently. Since, the mass flow rate of air is dependent on air density. The more dense the air is, the higher the mass flow rate is. If such an effect is not taken into account by using mass flow sampling, the resulting flux will be biased. The bias is equivalent to having the wind speed measurement dependent on air density in EC.

### 3.4 STEA/EC flux intercomparison

The measured $CO_2$ fluxes using the STEA method in flow-through mode showed a good match with the reference EC fluxes (Fig. (8)).

The time series of measured $CO_2$ fluxes in Fig. (8 - a) shows that the STEA method was able to reproduce the daily dynamics of $CO_2$ flux. The estimated fluxes using the STEA method appear to have less spikes and smoother in general, this is likely due to the smoothing effect of buffer volumes and the lower sensitivity of the closed path gas analyzer to rain and high humidity.

The mean diurnal cycle estimates from the two methods shown in Fig.(8 - b) match very well. However, a small time shift can be observed on the mean diurnal cycle a result of the phase shift introduced by the low-pass filtering effect of the buffer volumes.

The regression results shown in Fig. (8 - c) show that the measured $CO_2$ fluxes using the STEA method in flow-through mode have a very good agreement with the reference EC fluxes. The magnitude of STEA fluxes was comparable to EC fluxes (ODR slope = 1.05). This indicates that the STEA method does not introduce systematic error to the fluxes. However, the coefficient of determination $R^2$ was 0.87, which indicates a 13% unexplained variance contributed by the uncertainty in the two estimates. We suggest three different mechanisms contributing to the observed uncertainty leading to the unexplained variance. First, the random sampling error arising from the stochasticity of turbulence (Hollinger and Richardson, 2005). The observed uncertainty from the two methods calculated as the standard deviation of the difference is $4.29\ \mu\mathrm{mol\,m^{-2}\,s^{-1}}$ this is comparable to reported value of $2.7\ \mu\mathrm{mol\,m^{-2}\,s^{-1}}$ using two tower estimates (Hollinger and Richardson, 2005). The errors also show heteroscedasticity with the error increasing along with the absolute magnitude of the flux, a similar behavior was observed by Hollinger and Richardson (2005) when comparing two tower estimates. The second source of uncertainty is the

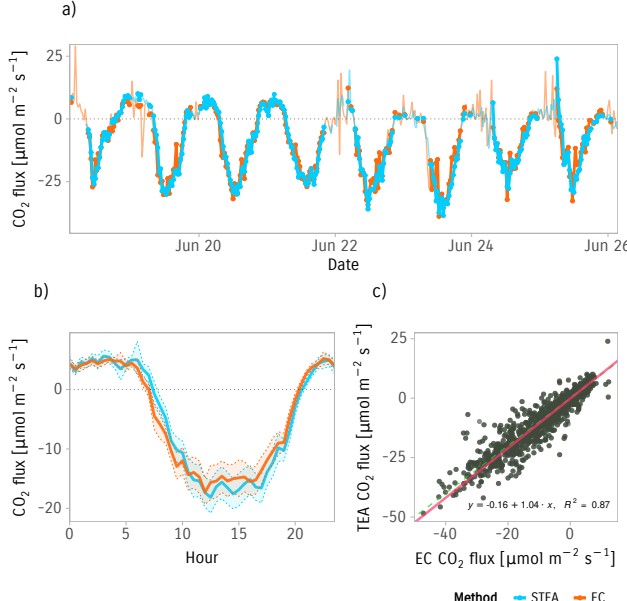

**Figure 8.** STEA and EC fluxes intercomparison. a) Time series of EC and STEA $CO_2$ fluxes for a subset period of 8 days. Points and thick lines indicate the averaging intervals used for comparison after filtering for quality. b) Mean diurnal cycle of $CO_2$ fluxes of STEA and EC. Bands are 95% confidence intervals of the mean calculated using nonparametric bootstrap. c) Scatter plot of STEA $CO_2$ fluxes against reference EC. The red line is the linear fit using the orthogonal distance regression (ODR). The dashed green line is a 1-to-1 line for reference.

use of different gas analyzers. Polonik et al. (2019) compared five different analyzers for measuring $CO_2$ fluxes. They showed that the root-mean-square error (RMSE) was in the range of 1 to $3.35\,\mathrm{\mu mol\,m^{-2}\,s^{-1}}$ depending on the analyzer type and the spectral correction method applied. Our results have an RMSE value of $4.3\,\mathrm{\mu mol\,m^{-2}\,s^{-1}}$. The third source of uncertainty is due to the use of buffer volumes in the STEA method. Figure (10 - a) demonstrates the increase of scatter in the measured
fluxes due to the use of buffer volumes.

Finally, the different processing steps between the two methods can contribute to the uncertainty. In particular, the effects of time-lag compensation, spectral corrections, and statistical screening. We determined the combined effect of these treatments by calculating the EC flux with and without the treatments and found that the effect on the flux was negligible.

### 3.5   Effect of buffer volumes

Using fixed buffer volumes attenuates the signal. The effect of buffer volumes can be described as a low-pass first-order linear filter. Figure (9) shows the filter's magnitude and phase responses. The magnitude response $|H|$ plot shows how the magnitudes of different frequencies are attenuated, the smaller the dimensionless flow rate is, the larger the time constant is. Consequently, the attenuation is stronger.

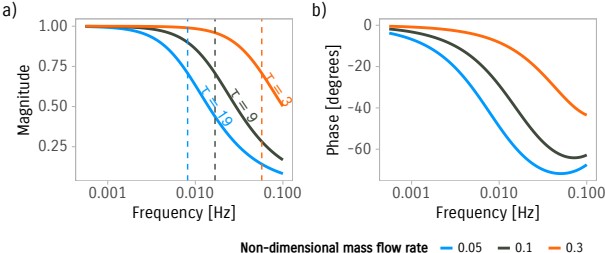

**Figure 9.** Frequency response for the first order linear filter used to model the buffer volumes for three different time constants. a) Magnitude response of the filter. Vertical dashed lines represent the cutoff frequency for the respective time constant. b) Phase response of the filter.

To understand the effect of buffer volumes use on the measured scalar concentration. We carried out a simulation on a surrogate signal generated from sonic temperature. The simulation showed the decline can reach up to 10% of the fluxes under operation ranges similar to those of our experiment (for $\tau = 11$ minutes) (Fig. 10). The empirical correction was consistently able to mitigate most of the attenuation when the filter properties are assumed to be constant, (i.e the flow rate needs to be constant for each short interval). This assumption was difficult to maintain using the 1-minute switching regime. The simulation showed the empirical correction for the buffer volumes worked best when the correction factor was obtained using a linear fit, as opposed to taking a ratio of the attenuated flux to the true flux for each averaging interval. The correction factor, in this case, is the reciprocal of the slope of the linear regression between the attenuated flux and the true flux. The correction factor calculated using Eq. (10) shows a good agreement between sensible heat flux and $CO_2$. However, the uncertainty of the correction factor increased with increasing buffer volume time constant. For our experiment, the average time constant for the first-order linear filter used to model the buffer volume was estimated to be $\tau = 700$ seconds. This value was used to simulate the loss on the fluxes using the sensible heat flux calculated from the sonic anemometer. The correction factor was obtained from the slope of the attenuated flux and was equal to $1.18$

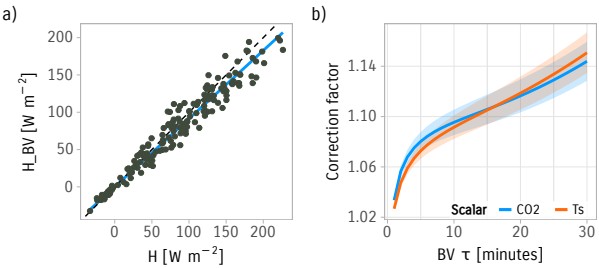

**Figure 10.** Empirical buffer volume correction. a) Effect of buffer volume attenuation on sensible heat flux with a time constant $\tau = 11$ minutes. The blue solid line is the linear fit between the two. b) Empirical correction factor for the effect of buffer volumes calculated as the reciprocal of the slope of attenuated flux for $CO_2$ and sensible heat flux. Bands are the estimated slope $\pm$ one standard error of the slope.





## 4  Conclusions

In this paper, we revised the theory of the true eddy accumulation method and extended the applicability of the method to non-ideal conditions where the mean vertical wind velocity during the averaging interval is not zero. The new generalized
equation allows estimating the scalar mean during the flux averaging interval and define conditions where the error in the flux is significant. We found that the error in the TEA flux is a function of the disparity of atmospheric transport and defined ways and conditions to constrain that error. We found that it is possible to achieve minimum bias in the TEA flux under most atmospheric conditions. We believe these results increase the confidence in using the TEA method for different atmospheric constituents and under a variety of atmospheric conditions.

Additionally, we proposed an alternative method for the measurement of ecosystem-level fluxes. The new method, referred to as short-time eddy accumulation (STEA), allows the sample accumulation to be carried out on shorter varying-length intervals. The method offers more flexibility than TEA and has many potential benefits including a better dynamic range and higher accuracy than the TEA method, and the ability to operate under a flow-through scheme using fixed buffer volumes. The flexibility introduced by the new method offers new ways to design eddy accumulation systems particularly suited for a specific
atmospheric constituent or a specific gas analyzer. For example, the accumulation time can be tailored to measure reactive species or to distribute the gas analyzer time to measure the fluxes at different heights.

Furthermore, we presented a prototype evaluation of the STEA method under the flow-through regime. We described the details of the system design and operation. We compared flux measurements from our new system against a reference EC system over a flat agricultural field. The fluxes from the two methods were in very good agreement. We highlighted the
importance of different processing and design aspects between the two methods and their potential effect on the fluxes.

Finally, we analyzed the effect of buffer volumes in the flow-through operational mode on the fluxes and proposed an empirical correction to correct for the underestimation resulting from the low-pass filtering behavior of the buffer volumes.

In summary, the generalized TEA equation and the new STEA method provide direct flux measurement methods that complement the state-of-the-art EC method. They extend the coverage of micrometeorological methods to new trace gases and
atmospheric constituents beyond the scope of the EC method.

*Code and data availability.* All data needed for producing the figures presented in the paper are provided at Emad and Siebicke (2021b). Scripts for producing the plots in the paper are available at Emad and Siebicke (2021a). Currently, drafts are accesible at: https://s.gwdg.de/R4Fdhg and https://s.gwdg.de/CZ4zXI.

## Appendix A:  Hicks and McMillen formulation

We show here how the TEA flux formula of Hicks and McMillen (1984), originally formulated under the assumption of $\bar{w} = 0$ is equivalent to using $(C_{acc}^{\uparrow} + C_{acc}^{\downarrow})/2$ as an estimate for $\bar{c}$ in the second term on the right hand side of Eq. (2).

We write the conditional expectation of $\bar{w}$ as





$$\overline{w} = \overline{\left( \overline{w | \text{sign}(w)} \right)} = \overline{|w^{\uparrow}|} \, \text{P}(w^{\uparrow}) - \overline{|w^{\downarrow}|} \, \text{P}(w^{\downarrow}) \tag{A1}$$

where $\text{sign}(w)$ is the sign of vertical wind velocity. $\text{P}(w^{\uparrow})$ and $\text{P}(w^{\downarrow})$ are the observed probabilities of the sign of $w$, which
equals the ratio of the time the wind is positive or negative to the total integration interval time.

$$\overline{w} = \overline{|w^{\uparrow}|} \frac{T^{\uparrow}_{avg}}{T_{avg}} - \overline{|w^{\downarrow}|} \frac{T^{\downarrow}_{avg}}{T_{avg}} \tag{A2}$$

and similarly

$$\overline{|w|} = \overline{|w^{\uparrow}|} \frac{T^{\uparrow}_{avg}}{T_{avg}} + \overline{|w^{\downarrow}|} \frac{T^{\downarrow}_{avg}}{T_{avg}} \tag{A3}$$

by substituting $\overline{|w|}/2$ with

$$\frac{\overline{|w|}}{2} = \overline{|w^{\uparrow}|} \frac{T^{\uparrow}_{avg}}{T_{avg}} - \frac{\overline{w}}{2} = \overline{|w^{\downarrow}|} \frac{T^{\downarrow}_{avg}}{T_{avg}} + \frac{\overline{w}}{2} \tag{A4}$$

we obtain

$$C^{\uparrow}_{acc}(\overline{w^{\uparrow}} \frac{T^{\uparrow}_{avg}}{T_{avg}} - \frac{\overline{w}}{2}) - C^{\downarrow}_{acc}(\overline{|w^{\downarrow}|} \frac{T^{\downarrow}_{avg}}{T_{avg}} + \frac{\overline{w}}{2}) \tag{A5}$$

After rearrangement and simplification we get to

$$F_{\text{EA}} = \overline{cw} - \overline{w} \left( \frac{C^{+}_{acc} + C^{-}_{acc}}{2} \right) \tag{A6}$$

When Eq. (A6) is compared with Eq. (2), it is clear that the term $\frac{C^{+}_{acc} + C^{-}_{acc}}{2}$ is used as an estimate for $\bar{c}$.

## Appendix B:  List of symbols

*Author contributions.*  AE developed the theory of the STEA method, the generalized TEA equation, and the empirical correction for the
effect of buffer volumes, implemented needed software, performed the experiment, analyzed the data, interpreted the results, and wrote the
manuscript. LS conceptualized the idea of flow-through eddy accumulation system, build the TEA system used in the experiment, planned
and supervised the experiment, provided feedback on the results, the analysis, and the manuscript.

*Competing interests.*  The authors declare that they have no competing interests.





**Table B1.** Symbols and subscripts with units

**Symbols**

| | | |
|---|---|---|
| $c$ | $\mathrm{mol\,m^{-3}}$ | Molar density of a scalar |
| $w$ | $\mathrm{m\,s^{-1}}$ | Vertical wind velocity |
| $T_{avg}$ | s | Flux averaging interval |
| $A$ | – | TEA sampling scaling factor |
| $V$ | $\mathrm{m^3}$ | Volume |
| $C$ | $\mathrm{mol\,m^{-3}}$ | Mean concentration of accumulated samples |
| $\alpha_c$ | – | Transport asymmetry coefficient for scalar $c$ |
| $\rho$ | – | Corelation coefficient |
| $\dot{q}$ | – | Dimensionless mass flow rate |
| $\tau$ | s | Time constant of the buffer volume |
| $r_c$ | ppm | Mixing ratio in dry air for a scalar, $c$ |

**Subscripts**

| | |
|---|---|
| $acc$ | Accumulated samples |
| $\uparrow$ | Updraft buffer volume |
| $\downarrow$ | Downdraft buffer volume |
| $c$ | Atmospheric constituent |

*Acknowledgements.* We gratefully acknowledge the support of the Bioclimatology group, Alexander Knohl, University of Göttingen, in particular technical assistance by Justus Presse, Frank Tiedemann, Marek Peksa, Dietmar Fellert, and Edgar Tunsch. We thank Christian Brümmer, Jean-Pierre Delorme from the Thünen Institute for Agricultural Climate Protection, and Mathias Herbst from the Center for Agrometeorological Research of the German Meteorological Service (DWD) for facilitating the field work in Braunschweig. We further acknowledge Christian Markwitz for the fruitful discussions during the preparation of the manuscript and for reading and commenting on the manuscript. We thank Alexander Knohl, Nicolò Camarretta, Justus van Ramshorst, and Yannik Wardius for reading the manuscript and providing useful comments.

*Financial support.* The study was financially supported by the Ministry of Lower-Saxony for Science and Culture (MWK), by the European Research Council under the European Union's Horizon 2020 research and innovation programme (grant agreement no. 682512 - OXYFLUX), and by the Deutsche Forschungsgemeinschaft (INST 186/1118-1 FUGG).





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
