# Peer review of "Advances in the True Eddy Accumulation Method: New theory, implementation, and field results"

_Atmospheric Measurement Techniques, 2021_

## Author Comment (AC1)

**Authors' reply to reviewer comments 1**

Anas Emad and Lukas Siebicke

**March 27, 2022**

Review of the manuscript amt-2021-319 "Advances in the True Eddy Accumulation Method: New theory, implementation, and field results" by Emad and Siebicke (2021)

**General comment**

We thank the anonymous reviewer for his/her constructive, motivating, and detailed feedback on the manuscript. Please find below our point by point reply to the comments.

- Our answers and comments are in blue.
- Summary of changes to the manuscript are in orange.

As per the suggestion of the second reviewer, Christoph Thomas we have split the manuscript into two papers. Comments related to the second manuscript will be indicated by *second manuscript* at the beginning.

The manuscript was split into two related papers

- 1. True eddy accumulation Part 1: Solutions to the problem of non-vanishing mean vertical wind velocity
- 2. True eddy accumulation Part 2: Theory and experiment of the short-time eddy accumulation method

**General comment**

The authors address one of the aerodynamic micrometeorological methods for surface-atmosphere turbulent flux measurement, the eddy accumulation. They propose interesting improvements to the classical method, namely the division of the half-hour averaging period into shorter periods and an approach to account for non-zero vertical wind speed. These are intended to reduce the uncertainties in the calculated fluxes and adapt the eddy accumulation system to the emerging faster-response gas analyzers. A theoretical framework explaining the basics of the method and the substantiating the proposed developments is well presented. The success of the modified eddy accumulation is apparent from the close similarity of the fluxes with the estimates from a collocated eddy-covariance system. I evaluate this work as a high quality contribution, which will certainly be of interest to the micrometeorological and flux measurement communities.

My criticisms are related to the treatment of the nighttime periods, the importance of different crops planted on either side of the EC tower leading to step-change in aerodynamic roughness, the actual extent of the vertical wind problem at the measuremet site since it is very flat, and the insufficient evidence of STEA technique superiority over the classical TEA method.

I recommend acceptance of the manuscript after the below points will be addressed.

**Detailed** comments**

Vertical wind more of a problem in sites with complex topography – this uncertainty needs to be discussed.

We agree the problem is larger in more complex topography, we have emphasized this in the introduction. However, although our experiment was conducted in an ideal site, we don't think this limits the generality of the results.

The relevancy to complex sites was stressed. A new sentence added: "The magnitude of residual nonzero mean vertical velocity is larger in complex sites (Rannik et al, 2020)"

48: I am at a loss as to what kind of error is implied – mean random error of the 30-min averaging periods, mean systematic error, error on the annual cumulative sum, ...? It would be good to clarify how big the problem you are solving really is.

The error is the mean systematic bias of several 30-min averaging intervals.

Changes:

Changed to mean systematic error.

94-95: the local topographical slope should be among the factors

**Changes:**

The topographical slope added to the factors as suggested.

100: "realizations" is unclear, perhaps explain that this means the need to record the fluctuations of scalar concentration at a frequency sufficient to represent the individual flux-transporting eddies, i.e. at frequencies higher than 1 Hz. Please specify how long an averaging period (or range) you mean at line 101.

**Changes:**

The sentence changed as suggested. And an example of the length of averaging period was added.

115-116 change "maps" for "ensures the proportionality"

*Changes:* Changed as suggested.

231-232: alpha has already been defined.

*Changes:* Redundant symbols removed as suggested.

Section 2.4: it is not quite clear how the division into shorter periods improves the flux estimates (if it is expected to improve them?) Maybe an introductory sentence explaining this should be added.

**second manuscript**

The division into shorter intervals improves the practical implementation of the TEA system in terms of dynamic range and flexibility such as running the system in a continuous flow-through mode.

Changes:

An introductory sentence was added to clarify the motivation for the division into shorter periods.

Section 2.6.1. The site description is missing important details on what crops were planted within the EC footprint during the measurement period, how tall the plants were etc. The photograph reveals that there was a significant change in surface roughness just near the tower, effectively dividing the EC flux source area into two contrasting halves. This would have long-reaching consequences for the surface exchange in general, and the functioning of aerodynamic flux measurement techniques, even if the step-change in roughness creates no significant wall effects affecting the

vertical wind speed. Please clarify these points. My recommendation is that, in case the crops were as shown in the photo and it shows in the wind direction dependent quality criteria of EC fluxes, the following sections should treat the two sectors separately.

We thank the reviewer for raising this important issue. The included photograph was taken during the installation and does not reflect the situation during the measurement period.

**Changes:**

We added the necessary information about the planted crops in the field during the measurements and their heights.

Fig. 3 and its discussion in the text: I think the explanation of the exact method of choosing the representative concentration for each short period is missing. Is it the mean or median? Are there any QC indicators to reject the poor quality short periods, based on e.g. the properties of distribution of recorded concentrations?

**second manuscript**

The median was chosen as the representative value. Several quality checks related to the performance and statistical properties of the measured values. They were detailed in Section 2.8.2.

**Changes:**

Updated Fig. 3 description to indicate that the median is used and referenced the other quality checks used for data selection.

Sections 2.9-2.10: how were the nights treated? From my experience, stable periods lead to the generation of thermal decoupling at 0.5-4 m height in such short vegetation sites, cutting the aerodynamic flux measurement system off from the ground sources of scalars. However, as it is well known, moderately stable periods lead to biases in EC/TEA fluxes even at the absence of decoupling. Can you briefly discuss how the performance of modified TEA differs from EC, and how much night-time data you had to reject for the above reasons.

**second manuscript**

Night data were indeed the majority of excluded values. Only 33 % of averaging intervals were valid during night-time compared to 70% during day time.

The exclusion of night data data was based on quality flags of (Mauder and Foken 2004) that test for violations of stationarity assumption and integral turbulence statistics.

In terms of performance, since TEA uses the same wind information as EC the only advantage it has is the close path gas analyzer. From a theoretical perspective the two methods have the same performance during nighttime but due to the use of difference gas analyzers, EC fluxes seemed to have more spikes during night-time due to the sensitivity of the open-path gas analyzer to water condensation while TEA closed path-gas analyzer was more robust.

**Changes:**

We added more information about night-time data quality and the performance difference between the two methods.

Line 493 and whole section 3.1.1: you are saying that there is an upper limit on the absolute value of mean vertical wind speed, up to which the alpha coefficient correction is still effective. At the same time, as your experiments takes place in a very flat ground (ignoring the surface roughness changes which require clarification), that one is led to think that nonzero vertical wind speed is of less concern for this site than it is, say, for a mountain site in, or a flat site with a discontinuous canopy consisting of large trees. Please provide some

We agree, for this site, the correction is minimal. This paragraph was intended to offer an example of using this new formulation for defining conditions where the error in the flux becomes significant.

More context was added to clarify the use of the defined upper limit on  $\overline{w}$  as a condition to restrict the error in the flux below a certain threshold.

Figure 5: the neutral periods are not visible - covered by the other symbols?

Indeed, most of them were covered by other symbols.

**Changes:**

We changed the transparency and the color to improve the visibility of neutral periods.

**503-505: what range are you referring to?**

**second manuscript**

Were are referring here to the dynamic range of the sampling apparatus.

**Changes:**

We changed the sentence wording to clarify what is meant by the range.

Figure 6b: the representation of volume as area of the boxes is confusing the way it is currently presented; I think it's better to have the bats of the same width and only vary their length.

**second manuscript**

Both the width and the height are variables. The width of the boxes is the accumulation interval and the height is the average mass flow rate which is also variable in each short averaging interval, their product is the area of the rectangle which represents the accumulated volume that need to be constant for all short intervals.

**Changes:**

**We improved the description of Fig. 6 to indicate why the height and width are variables.**

Figure 8 and the related text: It would be important to add the flux calculated using the traditional eddy accumulation approaches, to show that STEA offers superior performance in terms of smaller deviation from EC.

**second manuscript**

The effect of nonzero vertical wind velocity was very small in our site (< 1.5%) due to the ideal topography of the site and the online rotations and processing of the wind measurements, therefore we chose not to add the uncorrected TEA fluxes to the figures. As for comparing STEA with traditional TEA, the improvements in STEA are with system sampling and implementation and would need a reference TEA system to show the difference, this can not be realized in post processing without making assumptions about the limitations of the sampling apparatus and the type and use of buffer volumes.

---

## Author Comment (AC2)

**Authors' reply to reviewer comments**

Anas Emad and Lukas Siebicke

**May 21, 2022**

For the manuscript: "Advances in the True Eddy Accumulation Method: New theory, implementation, and field results" by Emad & Siebicke, https://doi.org/10.5194/amt-2021-319

**General comment**

We thank Christoph Thomas for his constructive, motivating, and detailed feedback on the manuscript. Please find below our point-by-point reply to the comments.

- Our answers and comments are in blue.
- Summary of changes to the manuscript is in orange.

**Overview** statement:**

This paper summarizes recent advances in developing the True Eddy Accumulation (TEA) as the only viable alternative to the eddy covariance (EC) method for determining direct ecosystem-scale fluxes for gas and potentially (not addressed here) particulate species for which fast-response analyzers do not exist. The authors are experts in this field and have been leading the development of TEA, I believe they are the only team currently working on it. In this sense, I commend these advances and was eager to learn about the progress this technique has made moving from its infancy to a more refined, potentially operational stage. I do have, however, serious doubts that the manuscript can be published in its current or mildly revised form because its structure does not lend itself for easy reading and application by non-experts of the flux community, and therefore is unlikely to have the impact it deserves. I am convinced of the merit of the TEA method, the quality of the proposed advances, and the expertise of its authors. In my opinion the manuscript suffers from the following substantial issues in descending order of importance:

1. The one manuscript combines three papers with three different foci: a) modifying the original TEA equation to non-vanishing mean vertical wind conditions, b) combining multiple flux estimates at time scales smaller than the intended averaging time scale, the author term 'STEA', which offers obvious practical advantages, and c) addressing and quantifying the effect of fixed buffer volumes and spectral water cross sensitivity of the CRDS analyzer for flow-through TEA applications. While all foci are relevant for the TEA method in general, combining them into one manuscript is not only confusing the reader, but causes it to lose focus, and burying important results in a shear endless stream of information. I am opposed to least-publishable-unit papers, and like the idea of one-stop-shopping about TEA, but I believe it contains too many entangling aspects. It's a tough 29-page read.

We agree the manuscript was long in its original form and has many ideas that made it difficult to read. We split the manuscript after consulting with the editor into two papers. The first paper deals with the theory parts of TEA under non vanishing mean vertical wind velocity. The second paper will contain the field measurements of the STEA method and other aspects related to the system design and operation such as buffer volumes. We believe these aspects are more relevant to the field experiments.

**Changes:**

The manuscript was split into two related papers

- 1. True eddy accumulation Part 1: Solutions to the problem of non-vanishing mean vertical wind velocity
- 2. True eddy accumulation Part 2: Theory and experiment of the short-time eddy accumulation method
- 2. The desire to produce TEA fluxes most closely matching those from EC is fundamentally ill-posed, and only relevant in the infancy stage of the method to prove 'yes we can do it'. I sincerely hope we have moved beyond this stage. Interpreting EC fluxes as true ecosystem behavior (as opposed to studying turbulence only) suffers from many known issues, and often invokes strong simplifications to methods and measurements. Instead, the user of the exciting TEA method and reader of the manuscript is interested in quantifying the total flux (ie the left-hand side of Reynold's second postulate, their equation (2)). In this sense, finding a complicated mathematical method to subtract the advective transport (ie flux by the mean flow) from the TEA flux for non-vanishing mean vertical motions conditions in their equation (16) is ill-posed. At least the authors portrait it this way and motivate their correction. In fact, the right-hand side term in (16) may be a necessary correction when the mean vertical motion does not vanish, but it is sold as a TEA equivalent to the WPL correction, which physically is incorrect. To elaborate, I believe the authors misinterpret the existing and well-accepted WPL correction, which states that the simple turbulent covariance term needs to be corrected for i) flux contributions from the contraction-expansion argument of air parcels passing through the fixed sampling volume of a gas analzers, and ii) a finite non-zero mean vertical motion for moist-air flux.

We acknowledge our terminology might have caused some confusion about the motivation and nature of this correction. The term eddy covariance was used in the context of the general method of eddy covariance rather than the "eddy flux" or the "turbulent flux". Similarly, the "advective term" that requires removal is the biased advective term, not the WPL term.

While both TEA and EC are formulated to measure the ecosystem flux,  $\overline{wc}$ . It's well known, it's not possible to calculate the total ecosystem flux from the measured w and c, since any additional bias (offset) will not affect in the covariance term but will appear in the measured term  $\overline{wc}$  which can be several times larger than the flux.

This measured biased advective term  $\bar{w}\bar{c}$  is trivial to discard in EC by block averaging or other means (thus using EC as a reference). While in TEA the nature of the real time sampling does not allow easy removal of this term. Our motivation here is to investigate the size of the error in the flux due to this biased advective term and explore ways to minimize or correct it. In this regards, we do not think TEA has any potential advantage over EC for measuring the true ecosystem flux. Given a sufficiently fast gas analyzer, TEA has the same amount of information as EC does but with the additional constraint of the real-time requirements. Therefore, the the true physical advective term "WPL term" it is not possible to be measured directly with TEA and need to be estimated by other means (similar to EC).

**Changes:**

- Better defined and unified terminology regarding the nature of the removed term across the paper.

- We adopted the term "biased advective term" to avoid confusion with WPL.
- 3. The use of the transport asymmetry coefficient  $\alpha_c$  has theoretical appeal but estimating it from available fastresponse scalar concentration/ sonic temperature for gas species, for which TEA is the ONLY available direct flux method, is flawed because of imperfect scalar-scalar similarity. Nobody requires TEA for estimating CO2 fluxes, for which results from a field study are shown, but fluxes for reactive gas species undergoing photochemical changes during eolian transport, and not just through surface sinks and sources (applicable to nonreactive scalar air temperature, carbon dioxide and water vapor concentrations, etc) are very unlikely to obey the invoked scalar-scalar similarity. They are many recent studies on this issue relating to the relaxed eddy accumulation (REA) group of methods, and now by introducing it to TEA through  $\alpha_c$  it gets contaminated too. Not sure if this is a true advance. I also struggle with the various definitions of  $\alpha_c$ , which may be inconsistent (see detailed comments below) and its physical interpretation.

We think the new formulation using the asymmetry coefficient,  $\alpha_c$  brings multiple advantages to TEA measurements even if no estimate for its value was made.

- 1. Most importantly, this formulation isolates the error in the flux and imposes an upper bound on its value without direct dependence on the scalar background concentration. Irrespective of the scalar, the upper bound for the error in the flux is  $\overline{w}/\overline{|w|}$  of the flux. This is valid when the values of  $|\alpha_c|$  are assumed to be below 1 for stationary turbulence. The arguments for this assumption are detailed below.
- 2. Unifying different formulas for TEA showing what is accounted for in each formula. e.g., using the mass difference as opposed to concentrations.
- 3. Possibility for removing or minimizing the error using the probabilistic approach or invoking scalar similarity which, we agree, neither is ideal but we think there is no way around this.

Regarding the different definitions of  $\alpha_c$ . We acknowledge that some details of how the different definitions link to each other was missing or not clear. In short,  $\alpha_c$  is defined as the ratio of the covariances

$$\alpha_c \equiv \frac{\overline{|w|'c'}}{\overline{w'c'}} \tag{1}$$

The link to the quadrants comes form the fact that  $|\overline{w}|'c'$  is equal to  $|\overline{w}'|c'$  when  $\overline{w} = 0$ , and a good approximation if  $\overline{w} \neq 0$ . We can write  $|\overline{w'}|c' = S1 + S4 - S2 - S3$ . We will add more details on this approximation and why it does not change the general conclusion. The expression in terms of quadrants allows  $\alpha$  to be written in terms of updraft flux (S1 + S4) and downdrafts flux (S2 + S3) as

$$\alpha_c = \frac{\mathrm{flux}^{\uparrow} - \mathrm{flux}^{\downarrow}}{\mathrm{flux}^{\uparrow} + \mathrm{flux}^{\downarrow}} \tag{2}$$

Here, updraft or downdraft fluxes refers to the wind direction and not the sign of the flux. From this equation, it is clear values of  $\alpha$  larger than 1, would indicate that updraft flux (S1 + S4) and downdraft flux (S2 + S3) have opposing signs. This would mean the quadrants will form a pattern similar to a butterfly which means for positive values of the wind the scalar wind are correlated and for negative values they are anti-correlated (or the other way around). This pattern is highly unlikely with stationary flows. The values of  $\alpha_c$  observed from our experiment support this analysis as seen below.

Stationarity test is done following (Foken and Wichura, 1996)

**Changes:**

We added additional details to how the different definitions relate to each other and a new plot for the observed values of  $\alpha$  for four scalars.

- 4. The style of the manuscript varies from section to section from equation-rich and full text explanations to bulletitem to-do list instructions. I realize that this diversity may be caused by alternating authors and could be helpful in in the field, but the manuscript would benefit from homogenization. Quite a few equations can easily be omitted to slim down and sharpen focus, but some require mor detailed derivations and explanations. I believe that the most important 1) could be addressed by splitting the one manuscript into several independent, but related parts (an option other boundary-layer micrometeorological journals offer), similar to a mini-series of publications. The handling editor will know and decide. Please find my additional detailed comments below.
- We have opted for splitting the manuscript into two related papers and made many editorial changes to improve the flow and structure of the manuscript. We think the diversity originates from the different topics covered. We hope splitting would offer a more homogeneous reading experience.

**Detailed comments (kept short for clarity)**

1. Line 32: please add 'out of the family of accumulation methods'.

**Changes:**

**Suggested change added**

2. Line 40: Please see general comment 2) above, I think it the desire to match the EC flux is flawed, but the true (total) flux should be in the center of our attention when conducting ecosystem studies.

**Changes:**

We rephrased this sentence to clearly reflect the aim of the research as mentioned in our reply to general comment 2.

3. Line 49: REA can lead to both under- and overestimation when compared to the EC flux, so I wouldn't call it just as a 'loss'. The sign of the change depends on the exact REA method used.

We agree loss is not an accurate characterization of the error.

```
Changes:
```

We changed it to systematic error.

5. Line 58: I think a first brief definition of  $\alpha_c$  is in order here, while I do have my doubts about its utility (see general comment 3) above). However, introducing it as a coefficient representing the disparity in the vertical transport between up and downdrafts is helpful.

We think the utility and definition should be more clear now as addressed in our reply to general comment 3.

6. Line 60ff: Here is the first break from 'sub-manuscript' a to b (see general comment 1) above).

Addressed in general comment 1.

7. Equation (3): It is uncommon to see the EC flux mathematically being defined in the time domain, but in reality it is (almost exclusively) done this way. I believe using number N instead of time increments is more universal, but is a matter of taste.

We agree this is a practical definition as we confirmed with the phrase "ensemble averages are estimated experimentally" The choice of dt was to be consistent with the definition as a continuous function.

8. Line 87: The statement about the merit of the WPL correction is incorrect, see general comment 2) above. Standard 1-D EC-based mass balance methods eliminate this term by setting  $\overline{w} \equiv 0ms^{-1}$  through rotation, but in reality it is a true physical term which requires accounting. This is what the WPL does. Flux estimates including the WPL term do therefore not 'correct' the covariance flux  $\overline{wtot}$ , but estimate the 'true' full transport  $\overline{wc}$ .

We agree that the characterizing of the WPL term as a correction is not accurate. We incorrectly used the term "eddy flux" to refer to the total ecosystem flux  $\overline{wc}$ . We corrected it to reflect that  $\overline{w'c'}$  is an approximation to the total ecosystem flux and the mean advective term,  $\overline{wc}$  in Eq. 2 is the "Web term".

Changes:

We used the correct terminology and restructured the paragraph to reflect the relation between the physical WPL term and the measured biased advective term.

9. Line 92: What do you mean by offset? Offsets are instrument-specific properties due to improper calibration or referencing. True physical non-zero vertical motions is a different phenomenon. Please be precise in terminology. I find this paragraph confusing, consider omission?

We used offset to refer to any non-turbulent component in w that will appear in the measured advective term  $\bar{c} \bar{w}$ . We think this paragraph is necessary to disentangle the relation between the physical advective WPL term and the biased measured one that we are trying to remove. We restructured this paragraph to make the distinction clear.

10. Line 101 and throughout: use often state 'computations at the end of the averaging interval', but more precise of 'for the averaging interval', since the decision of sampling into up- or down reservoir is done while sampling, and not just at the end.

Changes:

**Changed as suggested.**

11. Line 102: I recommend using a different symbol than  $T_{avg}$  for the averaging interval length, as it may be confused with temperature.

```
Changes:
```

The averaging interval was changed to  $\Delta t$  instead of  $T_{avg}$ .

12. Equation (6): I recommend removing it here, it does not lend anything to this section, and is only needed later in section 2.4 and Appendix A.

*Changes:* Equation (6) removed as suggested.

13. Equation (8) and (9) can be combined, since V is defined in equation (7). We agree

*Changes:* Equations were combined as suggested.

14. Line 33: I believe it may be helpful to point out that the magnitude of w needs to be taken first, then comes the averaging. If reversed, the statement does not apply.

**Changes:**

We agree, we added further explanation to emphasize this.

15. Line 136: See general comment 2) above. While your proposed correction takes a similar mathematical form, you do not discard the advective transport, but flux uncertainty because the assumption of zero mean-vertical velocity is not fulfilled. This difference is important!

This is correct. We have emphasized the correct use of terminology.

16. Equations (11) and (12): This is a key step, explain!

**Changes:**

We added further explanation. In particular, now we simplified the derivation by starting from c|w| without the need for the weighted mean. Eq. 9 in the new manuscript.

17. Line 164: add an 'and' between |w| and c.

*Changes:* Missing 'and' added as suggested.

18. Equation (16) and (17): rearranging is fine, but the utility is not obvious to me. Do we need (17)?

Equation 17 is the resulting corrected TEA flux, we think it is good to have although easily obtained from Eq. 16.

19. Equation (18): not needed, consider omitting.

```
Changes: Equation removed as suggested.
```

20. Equations (19) and (24): I struggle in combining these two different definitions: do you mean the magnitude of the terms  $flux \uparrow$  and  $flux \downarrow$ , or do these terms actually preserve their sign? To me, equation (24) is the inverse of (19). However, later in the results section (line 468 ff) you report values of  $\alpha$  between 0.2 and -0.18 for CO2, so

 $|\alpha|

**Changes:* Added variability of measurements to the text.**

33. Line 341ff: How do you correct for water vapor cross sensitivity as the wapor concentration remains identical? Maybe I do not understand the method correctly, please explain.

**second manuscript**

We have notice we missed some key information. This procedure utilizes the effect of air drying due to decompression to deliver a variable water vapor content. Starting from humid atmospheric air near saturation (RH ~ 90%, T=21). Air is sucked and compressed into the stainless steel buffer volumes to a pressure of (2.6 bar). The water partial pressure in the pressurized buffers will become higher than

saturation vapor pressure and water will precipitate leading to dryer air. Then, air is decompressed and forwarded to the analyzer. As the buffer pressure is reducing, water content will increase. Using this method we were able to modulate the water content in air from 6000 to 14000 ppm.

**Changes:**

We changed the text to reflect the used method more accurately.

34. Line 353: Please do not use the word 'deadband' here, as it is commonly used in the REA/ TEA community for discarding samples of small flux contribution (HREA) or small vertical velocity. This may lead to confusion, how about 'line flushing volume' or similar?

second manuscript We changed the deadband to "Flushing time"

35. Line 367: what is a dynamic flux unit? Please explain.

*second manuscript* We changed to the actual unit.

36. Section 2.8.4: Much of this is very technical and difficult to understand. These details distract from the true advances you propose, see general comment 1) above. For me, this belongs to submanuscript c).

*second manuscript* This section was shortened and improved.

37. Line 391: Iinear detrending, really? Why?

Detrending used here in the context of block averaging. We removed it to avoid confusion.

38. Section 2.10, line 400: you despike the time series of final computed fluxes excluding everything exceeding  $2\sigma$ ? This can only be justified by assuming a well behaved biological functioning of the carbon uptake, but not by turbulence transport. How much does that obscure true TEA flux uncertainty?

**second manuscript**

We agree despiking is not ideal as it could obscure valid flux measurements. However, for our dataset. Spikes were mostly due to problems with the open-path gas analyzer. Including the spikes in the analysis had very little effect on the comparison between EC and STEA, It improved the slope to 1.03 and slightly reduced  $R^2$  by 0.1.

Changes:

We relaxed the conditions for spike removals to exclude only the very obvious erroneous values. The new threshold (6 hour window, 2  $\sigma$ ) removes only 3 spikes from the whole dataset.

39. Section 2.11: so many filters and flags, I wonder what the TEA fluxes look like without applying all those....

**second manuscript**

The flags were necessary to exclude periods where one or both of the systems did not work well. In particular, rain and violations of the assumptions of the EC method accounted for the majority of excluded values which sometimes produced nonphysical fluxes (<  $100 \ \mu mol \ m^{-2} \ s^{-1}$ ). Therefore, the flags were only applied for the linear fit which is particularly sensitive to outliers to get a meaningful measure for comparing the two methods. No points were excluded from Fig. 7. which shows even without removing low quality fluxes, the fit is still reasonably good.

40. Section 3.1.1: please see general comment 3) above, and detailed comment t). I struggle with estimating numerical values for  $\alpha$  and relating them to your observations. Invoking scalar similarity weighs heavily on this TEA 'advance'. Have you investigated the scalar-scalar similarity for your field experiments? Even among conservative scalars it changes dramatically with time.

We have improved the definitions of  $\alpha$  as we detailed in our reply to comment 3. We hope it clarifies the numerical values of  $\alpha$

Changes:

The numerical values of  $\alpha$  were discussed in terms of the sign and magnitude.

41. Section 3.2: Much is repetition from section 2.8, please only report results and their discussion here.

second manuscript

Changes:

The earlier repetitions from section 2.8 were removed. We left the discussion on the advantages of the STEA method and how to achieve a flow-trough system with STEA as we think they highlight key findings of this study.

42. Line 525f: Please explain this to the reader, it is technical jargon.

**Changes:**

We added further explanation about the meaning of time-invariance and its relation to the buffer volumes model.

43. Line 539: Replace 'stable' by 'stationary'.

second manuscript Changed as proposed.

44. Line 545ff: Could the results be reversed here between long- and short-term rotation window?

**second manuscript**

We think the short-term rotation window will minimize the residual vertical wind velocity in most cases. However, we think these findings are site-specific and need to be evaluated in more complex sites.

45. Section 3.3.3: is this truly needed?

**second manuscript**

We think it is necessary to highlight the difference of WPL relevancy for flux calculation between EC and TEA. However, we agree it might be misplaced in results section, so we moved it to the methods section.

The section was shortened and moved to the methods section to

46. Line 578ff: Comparing the absolute uncertainties across studies is not useful as the site conditions and sampling methods are very different. This needs to be communicated in terms of relative flux estimate uncertainty.

**second manuscript**

We agree and have changed to using the random error estimate from our measurements and added a note about the limitation of RMSE for comparing different studies since we could not find a relative measure for comparing different gas analyzers.

**Changes:**

We used the random error calculated from our EC measurements to get an estimate about the contribution of the random sampling error to the uncertainty and added a sentence to highlight the limitation of RMSE for comparing cross-studies flux measurements.

47. Section 3.4: as this appears to be a key goal for the authors, I would like to see a comparison of their full new TEA method (equation 34) with the traditional TEA method (not including the proposed corrections), as well as the magnitude of the correction terms (second and third term in equation 16) separately. This will ultimately demonstrate the utility of the proposed corrections.

**second manuscript**

Under the specific conditions of our site, the third correction term using  $\alpha$  was small (less than 1.5% of the total flux). This was due to the flat topography and the use of online treatments to minimize the vertical wind and was highlighted in the results section but not shown in the figure comparing the two methods. However, not accounting for the second term and using the mass difference between updraft and downdraft, effectively measuring *overlinewc*, will result in a much larger error. The mean absolute error, in this case, is 170% of the turbulent flux.

Christoph Thomas, University of Bayreuth

Citation: https://doi.org/10.5194/amt-2021-319-RC2

---

## Referee Report (RR1)

**True eddy accumulation - Part 1: Solutions to the problem of non-vanishing mean vertical wind velocity**

The manuscript forms the 1$^{st}$ part of the revised duo of papers on the improvements to the true eddy accumulation method. This part deals with the correction for the non-zero mean vertical wind, which the authors conceptualize as $\alpha_c$. Several formulations of $\alpha_c$ are compared, deriving it from the standard vertical exchange equation, quadrant analysis and on analytical grounds. Convincing evidence of the effectiveness of this approach to analyze and reduce the uncertainty in TEA flux is presented. The previous comments have been taken into account.

**I therefore propose that this manuscript be accepted, given that the below minor changes will be made.**

Abstract, line 1: I think it's worth making this very specific – the TEA measures vertical turbulent exchange between the ecosystem and the atmosphere.

Line 7: the mention of advection comes too suddenly, I think this needs to be preceded by a sentence leading up to this idea.

The last sentence of Abstract should be revised, it doesn't read well.

Line 54: the sentence should be rewritten, I do not understand it.

111: "tilted coordinates" sounds too colloquial – perhaps rephrase as "non-alignment of the anemometer with the local topography" or something similar, depending what you mean. Likewise, the instruments you mean here are probably the anemometers.

114: need -> needs

Section 2.3. Since you are discussing the effect of coordinate rotation, you could mention the sector-wise planar fit method, which considerably improves the results compared with the use of a single plane.

Line 200: but deviations from Gaussian behavior may be expected given the various complications you listed above, most prominently the complex topography. Thus, the analytical solution should maybe be taken with caution.

244-252: I do not insist, but I think it would have been very illustrative to provide an extra figure showing time series of the three terms comprising Eq. 14 separately, so their relative size can be judged as it changes diurnally.

271: so is this due to the imperfect normality of the data?

Figure 2: the neutral cases are very difficult to see, consider changing the markers.

293-294: "indicates"

309: "poor proxy"